# EFFICIENT GRAPH NEURAL ARCHITECTURE SEARCH

## ABSTRACT

Recently, graph neural networks (GNN) have been demonstrated effective in various graph-based tasks. To obtain state-of-the-art (SOTA) data-specific GNN architectures, researchers turn to the neural architecture search (NAS) methods. However, it remains to be a challenging problem to conduct efficient architecture search for GNN. In this work, we present a novel framework for Efficient GrAph Neural architecture search (EGAN). By designing a novel and expressive search space, an efficient one-shot NAS method based on stochastic relaxation and natural gradient is proposed. Further, to enable architecture search in large graphs, a transfer learning paradigm is designed. Extensive experiments, including node-level and graph-level tasks, are conducted. The results show that the proposed EGAN can obtain SOTA data-specific architectures, and reduce the search cost by two orders of magnitude compared to existing NAS baselines.

## 1 INTRODUCTION

Recent years have witnessed the success of graph neural networks (GNN) (Gori et al., 2005; Battaglia et al., 2018) in various graph-based tasks, e.g., recommendation (Ying et al., 2018a), chemistry (Gilmer et al., 2017), circuit design (Zhang et al., 2019), subgraph counting (Liu et al., 2020), and SAT generation (You et al., 2019). To adapt to different graph-based tasks, various GNN models, e.g., GCN (Kipf & Welling, 2016), GAT (Veličković et al., 2018), or GIN (Xu et al., 2019), have been designed in the past five years. Most existing GNN models follow a neighborhood aggregation (or *message passing*) schema (Gilmer et al., 2017), as shown in the left part of Figure 1, which is that the representation of a node in a graph is learned by iteratively aggregating the features of its neighbors. Despite the broad applications of GNN models, researchers have to take efforts to design proper GNN architectures given different tasks by imposing different relational inductive biases (Battaglia et al., 2018). As pointed out by Battaglia et al. (2018), the GNN architectures can support one form of *combinatorial generalization* given different tasks, i.e., graphs. Then a natural and interesting question can be asked: *Can we automatically design state-of-the-art (SOTA) GNN architectures for graph-based tasks?* A straightforward solution is to adopt the neural architecture search (NAS) approaches, which have shown promising results in automatically designing architectures for convolutional neural networks (CNN) (Zoph & Le, 2017; Pham et al., 2018; Liu et al., 2019a; Tan & Le, 2019; You et al., 2020a).

However, it is nontrivial to adopt NAS to GNN. The first challenge is to define the search space. One can design a *dummy* search space to include as many as possible the related parameters, e.g., aggregation functions, number of layers, activation functions, etc., on top of the message passing framework (Eq. (1)), however, it leads to quite a large discrete space, for example, 315,000 possible GNN architectures are generated by including just 12 types of model parameters in You et al. (2020b)), which is challenging for any search algorithm. The second challenge is to design an effective and efficient search algorithm. In the literature, reinforcement learning (RL) based and evolutionary based algorithms have been explored for GNN architecture search (Gao et al., 2020; Zhou et al., 2019; Lai et al., 2020; Nunes & Pappa, 2020). However, they are inherently computationally expensive due to the stand-alone training manner. In the NAS literature, by adopting the weight sharing strategy, one-shot NAS methods are orders of magnitude more efficient than RL based ones (Pham et al., 2018; Liu et al., 2019a; Xie et al., 2019; Guo et al., 2019). However, the one-shot methods cannot be directly applied to the aforementioned dummy search space, since it remains unknown how to search for some model parameters like number of layers and activation functions by the weight sharing strategy. Therefore, it is a challenging problem to conduct effective and efficient architecture search for GNN.

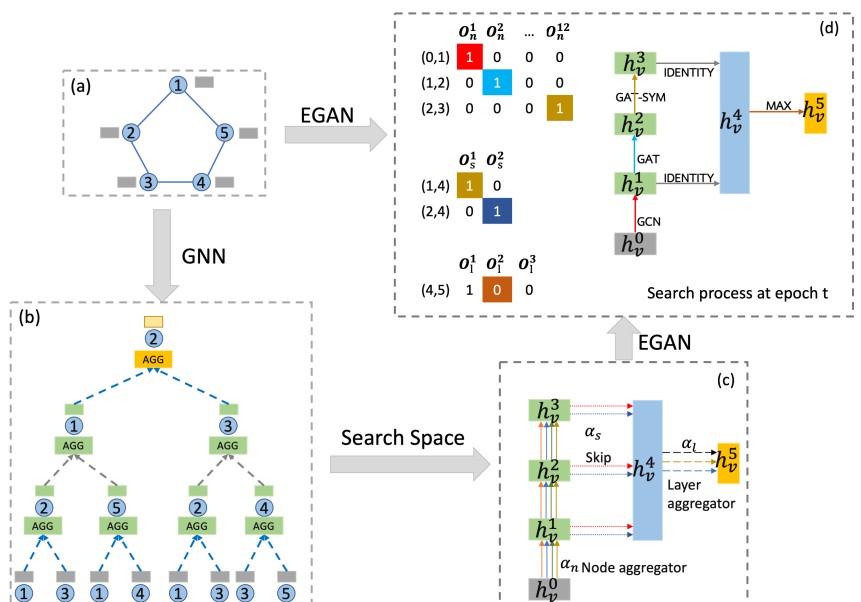

Figure 1: An illustrative example of a GNN model and the propsed EGAN (Best view in color). (a) An example graph with five nodes. The gray rectangle represents the input features of each node; (b) A typical 3-layer GNN model following the message passing neighborhood aggregation schema, which computes the embeddings of node "2"; (c) The DAG represents the search space for a 3-layer GNN, and $\boldsymbol{\alpha}_n, \boldsymbol{\alpha}_l, \boldsymbol{\alpha}_s$ represent, respectively, weight vectors for node aggregators, layer aggregators, and skip-connections in the corresponding edges. The rectangles denote the representations, out of which three green ones represent the hidden embeddings, gray ($\mathbf{h}_v^0$) and yellow ($\mathbf{h}_v^5$) ones represent the input and output embeddings, respectively, and blue one ($\mathbf{h}_v^4$) represent the set of output embeddings of three node aggregators for the layer aggregator. (d) At the $t$-th epoch, an architecture is sampled from $p(\mathbf{Z}_n), p(\mathbf{Z}_s), p(\mathbf{Z}_l)$, whose rows $\mathbf{Z}_{i,j}$ are one-hot random variable vector indicating masks multipled to edges $(i, j)$ in the DAG. Columns of these matrices represent the operations from $\mathcal{O}_n, \mathcal{O}_s, \mathcal{O}_l$.

In this work, we propose a novel framework, called EGAN (Efficient GrAph Neural architecture search), to automatically design SOTA GNN architectures. Motivated by two well-established works (Xu et al., 2019; Garg et al., 2020) that the expressive capabilities of GNN models highly rely on the properties of the aggregation functions, a novel search space consisting of node and layer aggregators is designed, which can emulate many popular GNN models. Then by representing the search space as a directed acyclic graph (DAG) (Figure 1(c)), we design a one-shot framework by using the stochastic relaxation and natural gradient method, which can optimize the architecture selection and model parameters in a differentiable manner. To enable architecture search in large graphs, we further design a transfer learning paradigm, which firstly constructs a proxy graph out of the large graph by keeping the properties, and then searches for GNN architectures in the proxy graph, finally transfers the searched architecture to the large graph. To demonstrate the effectiveness and efficiency of the proposed framework, we apply EGAN to various tasks, from node-level to graph-level ones. The experimental results on ten different datasets show that EGAN can obtain SOTA data-specific architectures for different tasks, and at the same time, reduce the search cost by two orders of magnitude. Moreover, the transfer learning paradigm, to the best of our knowledge, is the first framework to enable architecture search in large graphs.

**Notations.** Let $\mathcal{G} = (\mathcal{V}, \mathcal{E})$ be a simple graph with node features $\mathbf{X} \in \mathbb{R}^{N \times d}$, where $\mathcal{V}$ and $\mathcal{E}$ represent the node and edge sets, respectively. $N$ represents the number of nodes and $d$ is the dimension of node features. We use $N(v)$ to represent the first-order neighbors of a node $v$ in $\mathcal{G}$, i.e., $N(v) = \{u \in \mathcal{V} | (v, u) \in \mathcal{E}\}$. In the literature, we also create a new set $\widetilde{N}(v)$ is the neighbor set including itself, i.e., $\widetilde{N}(v) = \{v\} \cup \{u \in \mathcal{V} | (v, u) \in \mathcal{E}\}$.

## 2 RELATED WORKS

GNN was first proposed in (Gori et al., 2005) and in the past five years different GNN models (Kipf & Welling, 2016; Hamilton et al., 2017; Veličković et al., 2018; Gao et al., 2018; Battaglia et al.,

2018; Xu et al., 2019; 2018; Liu et al., 2019b; Abu-El-Haija et al., 2019; Wu et al., 2019; Zeng et al., 2020; Zhao & Akoglu, 2020; Rong et al., 2020) have been designed, and they rely on a neighborhood aggregation (or *message passing*) schema (Gilmer et al., 2017), which learns the representation of a given node by iteratively aggregating the hidden features ("message") of its neighbors. Besides, in Xu et al. (2018); Chen et al. (2020), the design of residual networks (He et al., 2016a;b) are incorporated into existing message passing GNN models. Battaglia et al. (2018) pointed out that the GNN architectures can provide one form of combinatorial generalization for graph-based tasks, and Xu et al. (2019); Garg et al. (2020) further show that the expressive capability of existing GNN models is upper bounded by the well-known Weisfeiler-Lehman (WL) test. It will be an interesting question to explore GNN architectures with better combinatorial generalization, thus neural architecture search (NAS) can be a worthwhile approach for this consideration.

NAS (Baker et al., 2017; Zoph & Le, 2017; Elsken et al., 2018) aims to automatically find SOTA architectures beyond human-designed ones, which have shown promising results in architecture design for CNN and Recurrent Neural Network (RNN) (Liu et al., 2019a; Zoph et al., 2018; Tan & Le, 2019). Existing NAS approaches can be roughly categorized into two groups according to search methods (Bender et al., 2018): i.e. the stand-alone and one-shot ones. The former ones tend to obtain the SOTA architecture from training thousands of architectures from scratch, including reinforcement learning (RL) based (Baker et al., 2017; Zoph & Le, 2017) and evolutionary-based ones (Real et al., 2019), while the latter ones tend to train a *supernet* containing thousands of architectures based on the weight sharing strategy, and then extract a submodel as the SOTA architecture at the end of the search phase (Pham et al., 2018; Bender et al., 2018; Liu et al., 2019a; Xie et al., 2019). The difference of training paradigms leads to that one-shot NAS methods tend to be orders of magnitude more efficient than the RL based ones.

Recently, there are several works on architecture search for GNN, e.g., RL based ones (Gao et al., 2020; Zhou et al., 2019; Lai et al., 2020), and evolutionary-based ones (Nunes & Pappa, 2020; Jiang & Balaprakash, 2020), thus all existing works are computationally expensive. Franceschi et al. (2019) proposes to jointly learn the edge probabilities and the parameters of GCN given a graph, thus orthogonal to our work. Besides, Peng et al. (2020) proposes a one-shot NAS method for GCN architectures in the human action recognition task. In Li et al. (2020), a sequential greedy search method based on DARTS (Liu et al., 2019a) is proposed to search for GCN architectures, however, the tasks are more focused on vision related tasks, with only one dataset in conventional node classification task. In this work, to the best of our knowledge, for conventional node-level and graph-level classification tasks, we are the first to design a one-shot NAS method for GNN architecture search, which is thus, in nature, more efficient than existing NAS methods for GNN. In Appendix A.3, we give more detailed discussions about the comparisons between EGAN and more recent GNN models.

## 3 THE PROPOSED FRAMEWORK

### 3.1 THE DESIGN OF SEARCH SPACE

As introduced in Section 2, most existing GNN architectures rely on a message passing framework (Gilmer et al., 2017), which constitutes the backbone of the designed search space in this work. To be specific, a $K$-layer GNN can be written as follows: the $l$-th layer ($l = 1, \cdots, K$) updates $\mathbf{h}_v$ for each node $v$ by aggregating its neighborhood as

$$\mathbf{h}_v^{(l)} = \sigma(\mathbf{W}^{(l)} \cdot \Phi_n(\{\mathbf{h}_u^{(l-1)}, \forall u \in \widetilde{N}(v)\})), \tag{1}$$

where $\mathbf{h}_v^{(l)} \in \mathbb{R}^{d_l}$ represents the hidden features of a node $v$ learned by the $l$-th layer, and $d_l$ is the corresponding dimension. $\mathbf{W}^{(l)}$ is a trainable weight matrix shared by all nodes in the graph, and $\sigma$ is a non-linear activation function, e.g., a sigmoid or ReLU. $\Phi_n$ is the key component, i.e., a pre-defined aggregation function, which varies across different GNN models.

Thus a dummy search space is to include as many as possible related parameters in Eq. (1). However, it leads to a very large search space, making the search process very expensive. In this work, motivated by two well-established works (Xu et al., 2019; Garg et al., 2020), which show that the expressive capabilities of GNN models highly rely on the properties of aggregation functions, we propose to search for different aggregation functions by simplifying the dummy search space. For other parameters, we do simple tuning in the re-training stage, which is also a standard practice in

existing NAS methods (Liu et al., 2019a; Xie et al., 2019). Then the first component of the proposed search space is the *node aggregators*, which consists of existing GNN models. To improve the expressive capability, we add the other component, *layer aggregators*, to combine the outputs of node aggregator in all layers, which have been demonstrated effective in JK-Network (Xu et al., 2018). Then we introduce the proposed search space, as shown in Figure 1(c), in the following:

- **Node aggregators**: We choose 12 node aggregators based on popular GNN models, and they are presented in Table 7 in Appendix A.1. The node aggregator set is denoted by $\mathcal{O}_n$.
- **Layer aggregators**: We choose 3 layer aggregators as shown in Table 7 in Appendix A.1. Besides, we have two more operations, IDENTITY and ZERO, related to skip-connections. Instead of requiring skip-connections between all intermediate layers and the final layer in JK-Network, in this work, we generalize this option by proposing to search for the existence of skip-connections between each intermediate layer and the last layer. To connect, we choose IDENTITY, and ZERO otherwise. The layer aggregator set is denoted by $\mathcal{O}_l$ and the skip operation set by $\mathcal{O}_s$.

To further guarantee that $K$-hop neighborhood can always be accessed, we add one more constraint that the output of the node aggregator in the last layer should always be used as the input of the layer aggregator, thus for a $K$-layer GNN architecture, we need to search $K-1$ IDENTITY or ZERO for the skip-connection options.

## 3.2 DIFFERENTIABLE ARCHITECTURE SEARCH

Following existing NAS works (Liu et al., 2019a; Xie et al., 2019), we represent the search space by a directed acyclic graph (DAG), as shown in Figure 1(c), where nodes represent embeddings, and edges represent operations between the two end nodes. Then the intermediate nodes are

$$x_j = \sum\nolimits_{i<j} \tilde{\mathbf{O}}_{i,j}(x_i), \tag{2}$$

where $\tilde{\mathbf{O}}_{i,j}$ is the selected operation at edge $(i,j)$. In our work, each edge corresponds to an operation in the search space, and we represent it with a distribution $p_{\boldsymbol{\alpha}}(\mathbf{Z})$, which generates the one-hot random variable $\mathbf{Z}_{i,j}$ multiplied by the operation edge $\mathbf{O}_{i,j}$ in the DAG. Then the intermediate nodes in each child graph are represented by

$$x_j = \sum\nolimits_{i<j} \tilde{\mathbf{O}}_{i,j}(x_i) = \sum\nolimits_{i<j} (\mathbf{Z}_{i,j})^{\mathbb{T}} \mathbf{O}_{i,j}(x_i). \tag{3}$$

Note that in our framework, as shown in Figure 1(c), for each node aggregator, the input is from the previous one, and for the layer aggregators, the input are from outputs of all node aggregators.

Following the setting in Zoph & Le (2017) and Gao et al. (2020), the objective of the framework is

$$\mathbb{E}_{\mathbf{Z}\sim p_{\alpha}(\mathbf{Z})}\big[R(\mathbf{Z})\big] = \mathbb{E}_{\mathbf{Z}\sim p_{\alpha}(\mathbf{Z})}\big[\mathcal{L}_{\mathbf{W}}(\mathbf{Z})\big], \tag{4}$$

where $R(\mathbf{Z})$ represents the reward, which is defined by training loss $\mathcal{L}_{\mathbf{W}}(\mathbf{Z})$ in our framework. $\mathbf{W}$ represents the model parameters. In the GNN literature, node-level or graph-level classification tasks are commonly used, thus the cross-entropy loss is chosen, leading to a differentiable function of $L_{\mathbf{W}}(\mathbf{Z})$. To make use of the differentiable nature of $L_{\mathbf{W}}(\mathbf{Z})$, we design a differentiable search method to optimize Eq. (4). To be specific, we use the Gumbel-Softmax (Maddison et al., 2017; Xie et al., 2019; Noy et al., 2020) to relax the discrete architecture distribution to be continuous and differentiable with the reparameterization trick:

$$\mathbf{Z}_{i,j}^k = f_{\alpha_{i,j}}(\mathbf{G}_{i,j}^k) = \exp((\log \boldsymbol{\alpha}_{i,j}^k + \mathbf{G}_{i,j}^k)/\lambda) / \sum\nolimits_{l=0}^{n} \exp((\log \boldsymbol{\alpha}_{i,j}^l + \mathbf{G}_{i,j}^l)/\lambda), \tag{5}$$

where $\mathbf{Z}_{i,j}$ is the softened one-hot random variable for operation selection at edge $(i,j)$, and $\mathbf{G}_{i,j}^k = -\log(-\log(\mathbf{U}_{i,j}^k))$ is the $k$-th Gumbel random variable, $\mathbf{U}_{i,j}^k$ is a uniform random variable. $\boldsymbol{\alpha}_{i,j}$ is the architecture parameter. $\lambda$ is the temperature of the softmax, which is steadily annealed to be close to zero (Xie et al., 2019; Noy et al., 2020). Then we can use gradient descent methods to optimize the operation parameters and model parameters together in an end-to-end manner. The gradients are given in Appendix A.2.

To improve the search efficiency, we further design an adaptive stochastic natural gradient method to update the architecture parameters in an end-to-end manner following Akimoto et al. (2019). To be specific, the update of $\boldsymbol{\alpha}$ at the $m$-th iteration is given as:

$$\boldsymbol{\alpha}^{m+1} = \boldsymbol{\alpha}^m - \rho \mathbf{H}^{-1} \nabla_{\boldsymbol{\alpha}} \mathcal{L}, \tag{6}$$

where $\rho$ is the step size. $\mathbf{H}$ is the Fisher matrix, which can be computed as $\mathbf{H} = \mathbb{E}_{p_{\boldsymbol{\alpha}^m}}[\bar{p}_{\boldsymbol{\alpha}}(\mathbf{Z})\bar{p}_{\boldsymbol{\alpha}}(\mathbf{Z})^{\mathbb{T}}]$ with $\bar{p}_{\boldsymbol{\alpha}}(\mathbf{Z}) := \nabla \log p_{\boldsymbol{\alpha}}(\mathbf{Z})$.

After the searching process terminates, we derive the final architecture by retaining the edge with the largest weight, which is the same as existing DARTS (Liu et al., 2019a) and SNAS (Xie et al., 2019). To make the final results more robust, the search process is executed 5 times with different random seeds, thus 5 architectures are obtained at the end of the search phase. Then the 5 architectures are re-trained from scratch with some hyperparameters tuning on the validation set, and the one with the best validation accuracy is returned as the final architecture.

### 3.3 Transfer learning paradigm

As introduced in Hamilton et al. (2017); Jia et al. (2020), when training GNN in large graphs, in each batch, the time and memory cost increases exponentially w.r.t. K, i.e., the number of GNN layers, with the worst cases of $O(|\mathcal{V}|)$. Obviously, it is extremely expensive in large graphs for any GNN model. The situation becomes more severe when conducting architecture search in large graphs since we are training a supernet emulating various GNN models. Therefore, it tends to be infeasible to directly search for architectures in large graphs. Motivated by transferring searched blocks and cells in CNN architectures from small to large data sets in the NAS literature (Zoph et al., 2018; Tan & Le, 2019), we propose to address the above problem by transfer learning (Pan & Yang, 2009).

The core idea of the transferable architecture search is to find a small proxy graph $\mathcal{G}_{\text{proxy}}$ (the source), then search in the proxy graph, finally tune the searched architecture $\{\boldsymbol{\alpha}_n, \boldsymbol{\alpha}_s, \boldsymbol{\alpha}_l\}$ in the large graph $\mathcal{G}$ (the target). However, in order to make the architecture transfer feasible, we need to make the proxy graph sharing the same property distribution with the original graph (Pan & Yang, 2009). Since the properties vary across different graphs, it is not suitable to transfer across different datasets, like that from CIFAR-10 to ImageNet for image classification (Zoph et al., 2018). Thus, we propose to sample a smaller graph from the original one, and then apply the transfer paradigm. Many distribution-preserving sampling schema have been proposed in an established work (Leskovec & Faloutsos, 2006), e.g., random sampling by node or edge, or sampling by PageRank. In this work, we adopt the Random PageRank Node (RPN) sampling method in Leskovec & Faloutsos (2006), which is empirically demonstrated to be able to preserve the properties by sampling not less than 15% nodes from the original graph. In Section 4.2.2, the experimental results shows that this transfer paradigm empirically works well.

Table 1: Comparisons between existing NAS methods for GNN and the proposed EGAN.

| | Search space | | Search | Able to run |
|---|---|---|---|---|
| | Node agg | Layer agg | Algorithm | in large graphs |
| GraphNAS (Gao et al., 2020) | $\sqrt{}$ | $\times$ | RL | $\times$ |
| Auto-GNN (Zhou et al., 2019) | $\sqrt{}$ | $\times$ | RL | $\times$ |
| Policy-GNN (Lai et al., 2020) | $\sqrt{}$ | $\times$ | RL | $\times$ |
| Nunes & Pappa (2020) | $\sqrt{}$ | $\times$ | Evolutionary | $\times$ |
| Jiang & Balaprakash (2020) | $\sqrt{}$ | $\times$ | Evolutionary | $\times$ |
| EGAN | $\sqrt{}$ | $\sqrt{}$ | Differentiable | $\sqrt{}$ |

### 3.4 Comparisons with existing NAS methods for GNN

In this section, as shown in Table 1, we emphasize the advantages of EGAN in the following:

• In terms of the search space, EGAN can emulate more GNN models than existing methods. Moreover, by only focusing on the "aggregation function", the total size of the search space is smaller than those of the previous methods, which also contributes to the efficiency improvements of EGAN.

• In terms of the search algorithm, the one-shot nature of EGAN makes it much more efficient than stand-alone methods, e.g. GraphNAS.

• The transfer paradigm of EGAN makes it feasible to conduct architecture search in large graphs.

Therefore, the advantages of EGAN over existing NAS methods are evident, especially the efficiency.

## 4 Experiments

In this section, we conduct extensive experiments to demonstrate the effectiveness and efficiency of the proposed EGAN, including node-level and graph-level tasks.

### 4.1 SETUP

**Datasets.** For node-level tasks, we have three settings: transductive, inductive, and transfer. The task is node classification on 8 datasets, which are given in Appendix A.4.1. For graph-level tasks, the task is whole graph classification on 2 datasets, which are given in Appendix A.5.

**Baselines** In general, we have two types of baselines: human-designed GNN models, and NAS methods. Details of baselines are given in Appendix A.4.2. Note that for NAS baselines in Table 1, we only use GraphNAS (Gao et al., 2020) and its variant using weight sharing (GraphNAS-WS). The search spaces of Auto-GNN (Zhou et al., 2019) and Nunes & Pappa (2020) are the same as GraphNAS, while their codes are not available. Jiang & Balaprakash (2020) is an concurrent work when we are preparing for this submission, and we will compare with it when the codes are available. For Policy-GNN (Lai et al., 2020), they work on searching for different numbers of layers per node in a selected GNN base model, i.e., GCN or GAT, thus it can be an orthogonal work to EGAN.

Table 2: Performance comparisons in transductive and inductive tasks, whose evaluation metrics are mean classification accuracy and Micro-F1, respectively. For the first five GNN models in the table, we present the better performance of each and its JK variants, and the detailed performance of GNN models and their JK variants are in Table 10 in Appendix A.4.5. Note that the performance of LGCN on CS and Computer is "-" due to the OOM (Out Of Memory) problem when running the released code in our GPUs. The results of Geom-GCN are copied from the original paper (Pei et al., 2020), since the data split ratio is the same to our experiments.

| | Methods | Transductive | | | | | Inductive |
| --- | --- | --- | --- | --- | --- | --- | --- |
| | | Cora | CiteSeer | PubMed | CS | Computer | PPI |
| Human-designed GNN | GCN | 0.877(0.012) | 0.771(0.014) | 0.878(0.004) | 0.949(0.003) | 0.912(0.007) | 0.934(0.001) |
| | GraphSAGE | 0.884(0.002) | 0.765(0.005) | 0.882(0.007) | 0.952(0.002) | 0.918(0.004) | 0.972(0.001) |
| | GAT | 0.873(0.009) | 0.753(0.013) | 0.867(0.006) | 0.934(0.004) | 0.918(0.005) | 0.978(0.001) |
| | GIN | 0.870(0.010) | 0.765(0.013) | 0.883(0.005) | 0.950(0.003) | 0.913(0.008) | 0.964(0.003) |
| | GeniePath | 0.878(0.012) | 0.759(0.014) | 0.882(0.004) | 0.926(0.004) | 0.883(0.007) | 0.964(0.001) |
| | LGCN | 0.869(0.008) | 0.754(0.022) | 0.875(0.001) | - | - | 0.772(0.002) |
| | Geom-GCN | 0.852* | 0.780* | **0.901*** | * | * | * |
| NAS methods | Random | 0.869(0.003) | 0.782(0.002) | 0.889(0.001) | 0.939(0.001) | 0.902(0.003) | 0.988(0.001) |
| | Bayesian | 0.858(0.003) | 0.765(0.002) | 0.884(0.001) | 0.943(0.001) | 0.908(0.001) | 0.990(0.001) |
| | GraphNAS | 0.884(0.007) | 0.776(0.006) | 0.890(0.002) | 0.928(0.002) | 0.900(0.003) | 0.970(0.013) |
| | GraphNAS-WS | 0.881(0.010) | 0.761(0.016) | 0.884(0.010) | 0.931(0.002) | 0.906(0.002) | 0.958(0.042) |
| | EGAN | **0.900(0.001)** | **0.786(0.011)** | 0.900(0.001) | **0.960(0.003)** | **0.920(0.003)** | **0.991(0.000)** |

### 4.2 PERFORMANCE COMPARISON

#### 4.2.1 PERFORMANCE COMPARISONS IN TRANSDUCTIVE AND INDUCTIVE TASKS

From Table 2, we can see that EGAN consistently obtain better or close performance compared to all baselines, which demonstrates the effectiveness of the proposed framework. In other words, with EGAN, we can obtain SOTA data-specific GNN architectures. When comparing EGAN with GraphNAS methods, the performance gain is evident. We attribute this to the expressive search space and the differentiable search algorithm. We further visualize the searched architectures in Figure 2, and we can see that the searched architectures vary per dataset. More figures can be checked in Figure 6 and 7 in Appendix A.6.3.

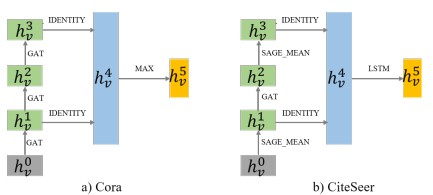

Figure 2: The searched architectures on Cora and CiteSeer.

#### 4.2.2 PERFORMANCE COMPARISONS IN TRANSFER TASKS

For the transfer learning experiments, we use two large graphs, i.e., Reddit and Arixv. As introduced in Section 3.3, we firstly sample two smaller graphs (15% nodes), and run EGAN in these two smaller graphs to obtain the optimal architectures, then transfer the searched architectures to the original graphs, finally report the test results in them. In terms of baselines, we only report the

results of those, which are able to, or reported to, run in the two large graphs, i.e., Reddit and Arxiv. More details are given in Appendix A.4.4.

The results are shown in Table 3, from which we can see that the improvement of search cost is in two orders of magnitude, from 3 GPU days[1] to less than a minute, which demonstrates the superiority of efficiency of the proposed transfer learning paradigm compared to direct architecture search in large graphs. Moreover, the performance of the transferred architectures is better or close to those of the baselines, which demonstrates the effectiveness of the transfer learning paradigm.

Table 3: Performance comparisons of the transfer task on Reddit and Arxiv, for which we use Micro-F1 and accuracy as evaluation metrics, respectively. Note that the OOM (Out Of Memory) problem occurs when running GraphNAS on these two datasets, thus we do not report its performance.

| | | Reddit | | | Arxiv | |
|---|---|---|---|---|---|---|
| | Method | Micro-F1 | Search cost (GPU hours) | Method | Accuracy | Search cost (GPU hours) |
| Human-designed GNN | GraphSAGE | 0.938(0.001) | - | GCN | **0.717(0.003)** | - |
| | GraphSAGE-JK | 0.942(0.001) | - | GraphSAGE | 0.715(0.003) | - |
| NAS methods | Random | 0.930(0.002) | 72 | Random | 0.691(0.005) | 72 |
| | Bayesian | 0.942(0.001) | 72 | Bayesian | 0.703(0.004) | 72 |
| | EGAN | **0.954(0.000)** | 0.021 | EGAN | 0.715(0.001) | 0.003 |

### 4.2.3 GRAPH-LEVEL TASKS

The results of the graph-level task are shown in Table 4, and we can see that the performance trending is similar to node-level tasks. The searched architectures are shown in Figure 7 in Appendix A.6.3, which also shows that they are data specific. Note that the global pooling method, or the readout function, in the whole graph learning can also be incorporated into the search space of EGAN, thus it can also be learned. We leave this for future work.

Taking into consideration the results of all tasks, the effectiveness of the proposed EGAN can be demonstrated.

Table 4: Performance comparisons on graph classification tasks. We show the mean test accuracy (with standard deviation) on these datasets. For the first five GNN models in the table, we present the better performance of each and its JK variants, and the detailed performance of GNN models and their JK variants are in Table 12 in Appendix A.5.3

| | Methods | D&D | PROTEINS |
|---|---|---|---|
| Human-designed GNN | GCN | 0.733(0.043) | 0.730(0.026) |
| | GraphSAGE | 0.734(0.029) | 0.734(0.041) |
| | GAT | 0.716(0.056) | 0.745(0.030) |
| | GIN | 0.733(0.033) | 0.737(0.048) |
| | GeniePath | 0.705(0.051) | 0.694(0.035) |
| | DiffPool | 0.779(0.045) | 0.738(0.040) |
| | SAGPOOL | 0.762(0.009) | 0.724(0.041) |
| | ASAP | 0.748(0.024) | 0.733(0.024) |
| NAS methods | Random | 0.742(0.043) | 0.731(0.031) |
| | Bayesian | 0.746(0.031) | 0.676(0.041) |
| | GraphNAS | 0.719(0.045) | 0.725(0.031) |
| | GraphNAS-WS | 0.758(0.044) | 0.752(0.025) |
| | EGAN | **0.779(0.034)** | **0.761(0.039)** |

### 4.3 SEARCH EFFICIENCY

In this section, we conduct some experiments to show the superiority of efficiency of EGAN over NAS baselines. And for simplicity, we only use the four commonly used datasets in the node-level task, i.e., Cora, CiteSeer, PubMed, and PPI.

Firstly, we record the running time of each method during the search phase, which represents the search cost of the NAS methods. The results are given in Table 5, from which we can see that the search cost of EGAN is two orders of magnitude smaller than those of NAS baselines.

Secondly, we show the trending of the test accuracy w.r.t. the running time of different methods during the search phase. In each epoch, we obtain the best model currently, and report the test accuracy after retraining it from scratch. The result of Cora is shown in Figure 3, from which we

---

[1]Note that we stop the search process after 3 days.

can observe that EGAN can obtain architectures with better performance more quickly than NAS baselines. More figures are shown in Figure 5 in Appendix A.6.2.

Taking these results into consideration, the efficiency advantage of EGAN is significant, which is mainly attributed to the one-shot training paradigm as introduced in Section 3.

|  | Transductive task | | | Inductive task |
|---|---|---|---|---|
|  | Cora | CiteSeer | PubMed | PPI |
| Random | 1,500 | 2,694 | 3,174 | 13,934 |
| Bayesian | 1,631 | 2,895 | 4,384 | 14,543 |
| GraphNAS | 3,240 | 3,665 | 5,917 | 15,940 |
| EGAN | 14 | 35 | 54 | 298 |

Table 5: The running time (s) of each method during the search phase.

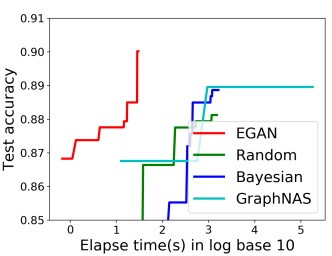

Figure 3: Test accuracy w.r.t. elapsed time on Cora.

## 4.4 ABLATION STUDY

In this section, we present two ablation studies on EGAN.

Firstly, we show the importance of the layer aggregators in the designed search space by running EGAN in the search space without the layer aggregators, and the results are shown in Table 6, from which we can see that the performance consistently drops on all datasets except Computer when removing the layer aggregators. This observation aligns with the results in JK-Network (Xu et al., 2018) that the performance of GNN models can be improved by adding an extra layer.

Secondly, we show the influence of $K$, i.e., the number of layers of GNN, in the search space, for which we conduct experiments with EGAN by varying layer $K \in \{1, 2, 3, 4, 5, 6\}$ and show the test accuracy in Figure 4. The results suggest that along with the increment of layers, the test accuracy may decrease. Considering the computational resources, 3-layer architecture is a good choice for the backbone of EGAN in our experiments.

|  | EGAN | |
|---|---|---|
|  | layer aggregators(w) | layer aggregators(w/o) |
| Cora | **0.900(0.001)** | 0.871(0.001) |
| CiteSeer | **0.759(0.000)** | 0.751(0.000) |
| PubMed | **0.900(0.001)** | 0.884(0.001) |
| CS | **0.947(0.001)** | 0.937(0.001) |
| Computer | 0.915(0.003) | **0.918(0.001)** |
| PPI | **0.990(0.000)** | 0.974(0.000) |
| DD | **0.779(0.034)** | 0.761(0.038) |
| Protein | **0.761(0.039)** | 0.758(0.034) |

Table 6: Performance comparisons of EGAN using different search spaces.

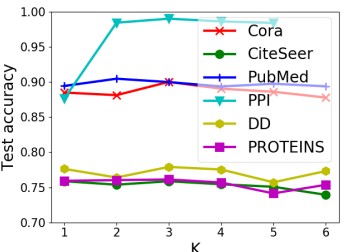

Figure 4: Test accuracy w.r.t. different $K$s.

## 5 CONCLUSION AND FUTURE WORK

In this paper, we propose EGAN, an effective and efficient framework for graph neural architecture search. By designing a novel and expressive search space, we propose a one-shot NAS framework by stochastic relaxation and natural gradient. Further, to enable architecture search in large graphs, we design a transfer learning paradigm. We conduct extensive experiments, including node-level and graph-level tasks, which demonstrates the effectiveness and efficiency of EGAN compared to various baselines.

Based on this work, we show that NAS approaches can obtain data-specific GNN architectures, which supports one form of combinatorial generalization for GNN models. For future work, we will explore more aspects regarding the combinatorial generalization of GNN models beyond the aggregation functions, like the construction of the graph, or the number of layers as done in Policy-GNN (Lai et al., 2020), as introduced in Battaglia et al. (2018).

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

# A APPENDIX

## A.1 DETAILS OF NODE AGGREGATORS

As introduced in Section 3.1, we have 12 types of node aggregators, which are based on well-known existing GNN models: GCN Kipf & Welling (2016), GraphSAGE Hamilton et al. (2017), GAT Veličković et al. (2018), GIN Xu et al. (2019), and GeniePath Liu et al. (2019b). Here we give key explanations to these node aggregators in Table 8. For more details, we refer readers to the original papers.

Table 7: The operations we use as node and layer aggregators for the search space of EGAN. Details of node aggregators are given in Table 8.

|  | Operations |
|---|---|
| $\mathcal{O}_n$ | SAGE-SUM, SAGE-MEAN, SAGE-MAX, SAGE-LSTM, GCN, GAT,GAT-SYM, GAT-COS, GAT-LINEAR, GAT-GEN-LINEAR, GIN, GeniePath |
| $\mathcal{O}_l$ | CONCAT, MAX, LSTM |
| $\mathcal{O}_s$ | IDENTITY, ZERO |

Table 8: More explanations to the node aggregators in Table 7.

| GNN models | Symbol in the paper | Key explanations |
|---|---|---|
| GCN (Kipf & Welling, 2016) | GCN | $F_N^l(v) = \sum_{u \in \widetilde{N}(v)} \left(\text{degree}(v) \cdot \text{degree}(u)\right)^{-1/2} \cdot \mathbf{h}_u^{l-1}$ |
| GraphSAGE (Hamilton et al., 2017) | SAGE-MEAN, SAGE-MAX, SAGE-SUM, SAGE-LSTM | Apply mean, max, sum, or LSTM operation to $\{\mathbf{h}_u \vert u \in \widetilde{N}(v)\}$. |
| GAT (Veličković et al., 2018) | GAT | Compute attention score: $\mathbf{e}_{uv}^{gat} = \text{Leaky\_ReLU}\left(\mathbf{a}\left[\mathbf{W}_u\mathbf{h}_u \vert\vert \mathbf{W}_v\mathbf{h}_v\right]\right)$. |
| | GAT-SYM | $\mathbf{e}_{uv}^{sys} = \mathbf{e}_{uv}^{gat} + \mathbf{e}_{vu}^{gat}$. |
| | GAT-COS | $\mathbf{e}_{uv}^{cos} = \langle \mathbf{W}_u\mathbf{h}_u, \mathbf{W}_v\mathbf{h}_v \rangle$. |
| | GAT-LINEAR | $\mathbf{e}_{uv}^{lin} = \tanh\left(\mathbf{W}_u\mathbf{h}_u + \mathbf{W}_v\mathbf{h}_v\right)$. |
| | GAT-GEN-LINEAR | $\mathbf{e}_{uv}^{gen-lin} = \mathbf{W}_G\tanh\left(\mathbf{W}_u\mathbf{h}_u + \mathbf{W}_v\mathbf{h}_v\right)$. |
| GIN (Xu et al., 2019) | GIN | $F_N^l(v) = \text{MLP}\left((1 + \epsilon^{l-1}) \cdot \mathbf{h}_v^{l-1} + \sum_{u \in N(v)} \mathbf{h}_u^{l-1}\right)$. |
| LGCN (Gao et al., 2018) | CNN | Use 1-D CNN as the aggregator. |
| GeniePath (Liu et al., 2019b) | GeniePath | Composition of GAT and LSTM-based aggregators |
| JK-Network (Xu et al., 2018) | | Depending on the base above GNN |

## A.2 GRADIENTS OF $\mathcal{L}$

Here we give the gradients of $\mathcal{L}_w(\mathbf{Z})$ w.r.t. the hidden embeddings, model parameters, and the architecture parameters, which is in the following:

$$
\begin{aligned}
\frac{\partial \mathcal{L}}{\partial x_j} &= \sum_{m>j} \frac{\partial \mathcal{L}}{\partial x_m} \mathbf{Z}_m^{\mathbb{T}} \frac{\partial \mathbf{O}_m(x_j)}{\partial x_j}, \\
\frac{\partial \mathcal{L}}{\partial \mathbf{W}_{i,j}^k} &= \frac{\partial \mathcal{L}}{\partial x_j} \mathbf{Z}_{i,j}^k \frac{\partial \mathbf{O}_{i,j}(x_i)}{\partial \mathbf{W}_{i,j}^k}, \\
\frac{\partial \mathcal{L}}{\partial \boldsymbol{\alpha}_{i,j}^k} &= \frac{\partial \mathcal{L}}{\partial x_j} \mathbf{O}_{i,j}^{\mathbb{T}}(x_i)(\boldsymbol{\delta}(k' - k) - \mathbf{Z}_{i,j})\mathbf{Z}_{i,j}^k \frac{1}{\lambda \boldsymbol{\alpha}_{i,j}^k}.
\end{aligned}
\tag{7}
$$

For full derivation, we refer readers to the appendix of Xie et al. (2019).

## A.3 DISCUSSION ABOUT RECENT GNN METHODS

Very recently there were some new GNN models proposed in the literature, e.g., MixHop (Abu-El-Haija et al., 2019), Geom-GCN (Pei et al., 2020), GraphSaint (Zeng et al., 2020), DropEdge (Rong et al., 2020), PariNorm (Zhao & Akoglu, 2020), PNA (Corso et al., 2020). We did not include all

these works into the search space of EGAN, since they can be regarded as orthogonal works of EGAN to the GNN literature, which means they are very likely to further increase the performance when integrating with EGAN.

As shown in Eq. (1), the embedding of a node $v$ in the $l$-th layer of a $K$-layer GNN is computed as:

$$\mathbf{h}_v^l = \sigma(\mathbf{W}^l \cdot \mathrm{AGG}_{\mathrm{node}}(\{\mathbf{h}_u^{l-1}, \forall u \in \widetilde{N}(v)\})).$$

From this computation process, we can summarize four key components of a GNN model: *aggregation function ($AGG_{node}$)*, *number of layers ($l$)*, *neighbors ($\widetilde{N}(v)$)*, and *hyperparameters ($\sigma$, dimension size, etc.)*, which decide the properties of a GNN model, e.g., the model capacity, expressive capability, and prediction performance.

To be specific, EGAN mainly focus on the aggregation functions, which affect the expressive capability of GNN models. GraphSaint mainly focuses on neighbors selection in each layer, thus the "neighbor explosion" problem can be addressed. Geom-GCN also focuses on neighbors selection, which constructs a novel neighborhood set in the continuous space, thus the structural information can be utilized. DropEdge mainly focuses on the depth of a GNN model, i.e., the number of layers, which can alleviate the over-smoothing problem with the increasing of the number of GNN layers. Besides the three works, there are more other works on the four key components, like MixHop (Abu-El-Haija et al., 2019) integrating neighbors of different hops in a GNN layer, or PairNorm (Zhao & Akoglu, 2020) working on the depth of a GNN models, and PNA (Corso et al., 2020) is a more recent GNN model, which proposes a composition of multiple aggregation functions in each GNN layer. Therefore, all these works can be integrated as a whole to improve each other. For example, the DropEdge or Geom-GCN methods can further help EGAN in constructing more powerful GNN models. With PNA, we can use our framework to help search for the combinations of multiple aggregation functions in a single GNN layer, or include the PNA aggregator to our search space to see whether it can help further enhance the final performance. This is what we mean "orthogonal" works of EGAN. Since we mainly focus on the aggregation functions in this work, we only compare the GNN variants with different aggregations functions.

Moreover, one purpose of this work is not to design the most powerful search space to include all aspects, but to demonstrate that the proposed EGAN, including the search space and search method, provides an alternative option to enhance GNN architecture search. We believe the application of NAS to GNN has unique values, and the proposed EGAN can benefit the GNN community.

## A.4 EXPERIMENT SETUP OF NODE-LEVEL TASKS

### A.4.1 DATASETS

Table 9: Dataset statistics of the datasets in the experiments. Here we use "CS", "Computer", as the abbreviations for Coauthor CS and Amazon Computers, respectively.

| | Transductive | | | | | Inductive | Transfer | |
| --- | --- | --- | --- | --- | --- | --- | --- | --- |
| | Cora | CiteSeer | PubMed | CS | Computer | PPI | Reddit | Arxiv |
| #nodes | 2,708 | 3,327 | 19,717 | 18,333 | 13,752 | 56,944 | 232,965 | 169,343 |
| #edges | 5,278 | 4,552 | 44,324 | 81,894 | 245,861 | 818,716 | 57,307,946 | 1,166,243 |
| #features | 1,433 | 3,703 | 500 | 6,805 | 767 | 121 | 602 | 128 |
| #classes | 7 | 6 | 3 | 15 | 10 | 50 | 50 | 40 |

**Transductive Setting** Only a subset of nodes in one graph are used as training data, and other nodes are used as validation and test data. For this setting, we use three benchmark dataset: Cora, CiteSeer, PubMed. They are all citation networks, provided by (Sen et al., 2008). Each node represents a paper, and each edge represents the citation relation between two papers. The datasets contain bag-of-words features for each paper (node), and the task is to classify papers into different subjects based on the citation networks.

Besides the three benchmark datasets, we use two more datasets: Coauthor CS and Amazon Computers, provided by (Shchur et al., 2018). Coauthor CS is a co-authorship graph where nodes are authors which are connected by an edge if they co-author a paper. Given paper keywords for each

author's paper as node features, the task is to map each author to its most active field of study. Amazon Computers is segments of the Amazon co-purchase graph where nodes represent goods which are linked by an edge if these goods are frequently bought together. Node features encode product reviews as bag-of-word feature vectors, and class labels are given by product category.

For all 5 datasets, We split the nodes in all graphs into 60%, 20%, 20% for training, validation, and test. For the transductive task, we use the classification accuracy as the evaluation metric.

**Inductive Setting** In this task, we use a number of graphs as training data, and other completely unseen graphs as validation/test data. For this setting, we use the PPI dataset, provided by (Hamilton et al., 2017), on which the task is to classify protein functions. PPI consists of 24 graphs, with each corresponding to a human tissue. Each node has positional gene sets, motif gene sets and immunological signatures as features and gene ontology sets as labels. 20 graphs are used for training, 2 graphs are used for validation and the rest for testing, respectively. For the inductive task, we use Micro-F1 as the evaluation metric.

**Transfer Setting** In this task, we use two datasets, Reddit and Arxiv, which are two orders of magnitude larger than Cora and CiterSeer in number of nodes, as shown in Table 9. The Reddit dataset is provided by Hamilton et al. (2017), and the task is to predict the community to which different Reddit posts belong. Reddit is an online discussion forum where users comment on different topics. Each node represents a post, and each edge represents a link between two posts, when they are commented by the same user. The dataset contains word vectors as node features. The graph is constructed from Reddit posts made in the month of September 2014, and we follow the same settings in the original paper (Hamilton et al., 2017), which uses the first 20 days for training and the remaining days for test (with 30% used for validation).

The Arxiv dataset is constructed based on the citation network between all papers, and we use the specific version of `ogbn-arxiv`, provided by a recent open graph benchmark (OGB) project (Hu et al., 2020), where the task is to predict the 40 subject areas of Arxiv CS papers, e.g., cs.AI, cs.LG. Each node (paper) has a 128-dimensional feature vector obtained by averaging the embedding of words in its title and abstract. And all papers are also associated with the year that the corresponding paper was published. The dataset is split by time. To be specific, papers published before 2017 are used as training data, while those in 2018 and 2019 are used, respectively, as validation and test set. For more details, we refer readers to Hu et al. (2020).

### A.4.2 COMPARED METHODS

We compare EGAN with two groups of state-of-the-art methods: human-designed GNN architectures and NAS methods for GNN.

**Human-designed GNNs.** We use the following popular GNN architectures:

- GCN (Kipf & Welling, 2016) proposes a sum aggregator normalized by the degrees of nodes.
- GraphSAGE (Hamilton et al., 2017) proposes scalable graph neural network with different aggregators: Mean, Sum, Max-Pool, LSTM.
- GAT (Veličković et al., 2018) proposes the attention aggregators, and it has different variants according to the attention functions: GAT, GAT-SYS, GAT-LINEAR, GAT-COS, GAT-GENERALIZED-LINEAR. The detail of these attention functions are given in (Gao et al., 2020).
- GIN (Xu et al., 2019) proposes to use Multi-layern Perceptron (MLP) as aggregators.
- LGCN (Gao et al., 2018) proposes to automatically select topK neighbors for each node, and use the 1-D regular convolutional operation as the aggregator.
- GeniePath (Liu et al., 2019b) proposes a composition of attention and LSTM-style aggregators, which can learn adaptive neighborhood for each node.
- Geom-GCN (Pei et al., 2020) propose a geometric bi-level aggregation schema over structure-aware neighbors in an continuous space and neighbors by adjacency matrix.

For models with variants, like different aggregators in GraphSAGE or different attention functions in GAT, we report the best performance across the variants. Besides, we extend the idea of JK-Network (Xu et al., 2018) in all models except for LGCN, and obtain 5 more variants: GCN-JK, GraphSAGE-JK, GAT-JK, GIN-JK, GeniePath-JK, which add an extra layer. In the experiments, we

only report the better performance of each GNN and its JK variant, which is denoted by the original name, as shown in Table 2, and 4, respectively.

For LGCN, we use the code released by the authors [2]. For other baselines, we use the popular open source library Pytorch Geometric (PyG) (Fey & Lenssen, 2019) [3] (Version: 1.6.0), which implements various GNN models. For all baselines, we train it from scratch with the obtained best hyperparameters on validation datasets, and get the test performance. We repeat this process for 5 times, and report the final mean accuracy with standard deviation.

**NAS methods for GNN.** We consider the following methods:

- Random search (denoted as "Random") is a simple baseline in NAS, which uniformly randomly samples architectures from the search space, and keeps track of the optimal architecture during the search process.

- Bayesian optimization[4] (denoted as "Bayesian") (Bergstra et al., 2011) is a popular sequential model-based global optimization method for hyper-parameter optimization, which uses tree-structured Parzen estimator as the metric for expected improvement.

- GraphNAS[5] (Gao et al., 2020), a NAS method for searching GNN architecture, which is based on reinforcement learning (Zoph & Le, 2017).

Random and Bayesian are searching on the designed search space of EGAN, where a GNN architecture is sampled from the search space, and trained till convergence to obtain the validation performance. 5000 models are sampled in total and the architecture with the best validation performance is trained from scratch, and do some hyperparameters tuning on the validation dataset, and obtain the test performance. For GraphNAS, we set the epoch of training the RL-based controller to 5000, and in each epoch, a GNN architecture is sampled, and trained for enough epochs ($600 \sim 1000$ depending on datasets), update the parameters of RL-based controller. In the end, we sample 10 architectures and collect the top 5 architectures that achieve the best validation accuracy. Then the best architecture is trained from scratch. Again, we do some hyperparameters tuning based on the validation dataset, and report the best test performance. Note that we repeat the re-training of the architecture for five times, and report the final mean accuracy with standard deviation.

Note that for human-designed GNN models and NAS methods, for fair comparison and good balance between efficiency and performance, we choose set the number of GNN layers to be 3, which is an empirically good choice in the literature (Veličković et al., 2018; Liu et al., 2019b).

### A.4.3   IMPLEMENTATION DETAILS OF EGAN

Our experiments are running with Pytorch (version 1.6.0) on a GPU 2080Ti (memory: 12GB, cuda version: 10.2). We implement EGAN on top of the building codes provided by PyG (version 1.6.0) and SNAS [6]. For all tasks, we run the search process for 5 times with different random seeds, and retrieve top-1 architecture each time. By collecting the best architecture out of the 5 top-1 architectures on validation datasets, we repeat 5 times the process in re-training the best one, fine-tuning hyperparameters on validation data, and reporting the test performance. Again, the final mean accuracy with standard deviations are reported.

In the training stage, we set the search epoch to 600 for all datasets except PPI (150), and the learning rate to 0.005, $L_2$ norm to 0.0005, dropout rate to 0.5. In the fine-tuning stage, each architecture is trained from scratch with 600 epochs.

### A.4.4   BASELINES IN TRANSFER LEARNING

On Reddit, we follow the same settings of GraphSAGE in the original paper (Hamilton et al., 2017), except that we use a 3-layer GNN as backbone to search, while GraphSAGE use 2-layer. Since the performance of Reddit is only reported in GraphSAGE (Hamilton et al., 2017) and JK-Network (Xu

---

[2]https://github.com/HongyangGao/LGCN

[3]https://github.com/rusty1s/pytorch_geometric

[4]https://github.com/hyperopt/hyperopt

[5]https://github.com/GraphNAS/GraphNAS

[6]https://github.com/SNAS-Series/SNAS-Series

et al., 2018), we use them as human-designed architectures. Note that the authors of JK-Network did not release their implementation, thus we use the implementation by the PyG framework. To keep it consistent, we also use the implementation for GraphSAGE by the PyG framework. For Arxiv, we follow the same setting in OGB project (Hu et al., 2020), where only two human-designed architectures, GCN and GraphSAGE, are tested, thus we use these two as human-designed architectures.

For NAS baselines, we use the same NAS approaches in transductive and inductive tasks: Random, Bayesian, GraphNAS, and GraphNAS-WS, and run them directly in the original graphs, i.e., Reddit and Arxiv. However, since the code of GraphNAS and GraphNAS-WS crashed due to *out of memory* error, we only report the performance of Random and Bayesian. Besides, we report the search cost in terms of GPU hours to compare the efficiency of different methods.

### A.4.5    PERFORMANCE COMPARISONS OF GNN MODELS AND JK VARIANTS

The detailed performance of GNN baselines and their JK variants in Section 4.2.1 are in Table 10.

Table 10: Performance comparisons between the five GNN models and their JK variants. For simplicity, we report the better one of each in Table 2.

|  | Transductive | | | | | Inductive |
|---|---|---|---|---|---|---|
|  | Cora | CiteSeer | PubMed | CS | Computer | PPI |
| GCN | 0.876(0.010) | 0.766(0.020) | 0.871(0.003) | 0.935(0.005) | 0.912(0.007) | 0.933(0.002) |
| GCN-JK | 0.877(0.012) | 0.771(0.014) | 0.878(0.004) | 0.949 (0.003) | 0.910(0.006) | 0.934(0.001) |
| GraphSAGE | 0.874(0.002) | 0.760(0.009) | 0.878(0.004) | 0.933(0.005) | 0.915(0.003) | 0.972(0.001) |
| GraphSAGE-JK | 0.884(0.002) | 0.765(0.005) | 0.882(0.007) | 0.952(0.002) | 0.918(0.004) | 0.972(0.001) |
| GAT | 0.872(0.016) | 0.752(0.014) | 0.857(0.007) | 0.929(0.003) | 0.917(0.006) | 0.978(0.001) |
| GAT-JK | 0.873(0.009) | 0.753(0.013) | 0.867(0.006) | 0.934(0.004) | 0.918(0.005) | 0.978(0.001) |
| GIN | 0.860(0.008) | 0.734(0.014) | 0.880(0.005) | 0.921(0.005) | 0.867(0.041) | 0.959(0.005) |
| GIN-JK | 0.870(0.010) | 0.765(0.013) | 0.883(0.005) | 0.950(0.003) | 0.913(0.008) | 0.964(0.003) |
| GeniePath | 0.867(0.012) | 0.759(0.014) | 0.880(0.004) | 0.926(0.004) | 0.861(0.008) | 0.953(0.000) |
| GeniePath-JK | 0.878(0.012) | 0.759(0.012) | 0.882(0.004) | 0.925(0.005) | 0.883(0.007) | 0.964(0.000) |

### A.5    EXPERIMENT SETUP OF GRAPH-LEVEL TASKS

### A.5.1    DATASETS

Table 11: The statistics of datasets in graph-level tasks.

| Dataset | Num. Graphs | Classes | Avg. Number of Nodes | Avg. Number of Edges |
|---|---|---|---|---|
| D&D | 1,178 | 2 | 384.32 | 715.66 |
| PROTEINS | 1,113 | 2 | 39.06 | 72.82 |

In this section, we evaluate EGAN on graph classification task two datasets: D&D and PROTEINS datasets, provided in Dobson & Doig (2003). These two datasets are both the protein graphs. In D&D dataset, nodes represent the amino acids and two nodes are connected iff the distance is less than 6 $\mathring{A}$. In PROTEINS dataset, nodes are secondary structure elements and edges represent nodes are in an amino acid or in a close 3D space. More information are shown in Table 11.

### A.5.2    BASELINES

Besides the GNN baselines in node-level tasks, we use three more methods, which use hierarchy pooling to learn whole graph representation given the embeddings of all nodes. DiffPool (Ying et al., 2018b), SAGPool (Lee et al., 2019) and ASAP (Ranjan et al., 2020) are latest methods based on hierarchical pooling schema, which learn node embeddings with node aggregators and coarsen graphs with pooling aggregators. The final graph embeddings are generated by a readout operation based on the final coarsend graph. DiffPool learns a soft assignment matrix for each node with any GNN methods, combine with entropy regularization and link prediction objective, so that the

coarsened graph can preserve as much information as possible. SAGPool learns node weights with attention mechanism and keeps the top-k nodes in pooling layer. In ASAP, it learns a soft cluster assignment matrix for each node with self-attention mechanism, and calculates the fitness score for each cluster and select top-k clusters.

For other methods, including GNN models in node-level tasks and NAS methods used in Table 4, to obtain the representation of a whole graph, we use the global sum pooling method at the end of retraining the derived architecture, i.e., the whole graph representation is obtained by the summation of the embeddings of all nodes. $\mathbf{z} = \sum_{i \in \mathcal{V}} \mathbf{h}_i^{(K)}$., $K$ is the number of GNN layers.

In this section, we use 10-fold cross-validation accuracy as the evaluation metric, and the implementation details are presented in A.4.3. After finding the best architecture and tuning the hyperparameters, we report the mean accuracy and standard deviations on 10 folds data.

In the search stage, we set the search epoch to 150, and the learning rate to 0.01, $L_2$ norm to 0.0005, dropout rate to 0.5. In the re-training stage, each architecture is trained from scratch with 100 epochs.

### A.5.3 PERFORMANCE COMPARISONS OF GNN MODELS AND JK VARIANTS

The detailed performance of GNN baselines and their JK variants in Section 4.2.3 are in Table 12.

Table 12: Performance comparisons between the five GNN models and their JK variants. For simplicity, we report the better one of each in Table 4.

|  | D&D | PROTEINS |
|---|---|---|
| GCN | 0.727(0.039) | 0.727(0.026) |
| GCN-JK | 0.733(0.043) | 0.730(0.026) |
| GraphSAGE | 0.734(0.029) | 0.734(0.031) |
| GraphSAGE-JK | 0.728(0.047) | 0.730(0.035) |
| GAT | 0.713(0.052) | 0.734(0.037) |
| GAT-JK | 0.716(0.056) | 0.745(0.030) |
| GIN | 0.732(0.030) | 0.736(0.028) |
| GIN-JK | 0.733(0.033) | 0.737(0.048) |
| GeniePath | 0.704(0.039) | 0.670(0.045) |
| GeniePath-JK | 0.705(0.051) | 0.694(0.035) |

### A.6 MORE EXPERIMENTAL RESULTS

Table 13: Performance comparisons of two search spaces on four benchmark datasets. We show the mean classification accuracy (with STD).

| Methods | Cora | CiteSeer | PubMed | PPI |
|---|---|---|---|---|
| GraphNAS | 0.884(0.007) | **0.776(0.006)** | 0.890(0.002) | 0.970(0.013) |
| GraphNAS-WS | 0.881(0.010) | 0.761(0.016) | 0.884(0.010) | 0.958(0.042) |
| GraphNAS (Our search space) | 0.883(0.002) | 0.771(0.006) | 0.888(0.001) | **0.989(0.001)** |
| GraphNAS-WS (Our search space) | **0.890(0.005)** | 0.770(0.007) | **0.894(0.001)** | 0.988(0.001) |

### A.6.1 THE ADVANTAGE OF THE PROPOSED SEARCH SPACE

In Section 3.1, we discuss the advantages of search space between EGAN and GraphNAS/Auto-GNN. In this section, we conduct experiments to further show the advantages. To be specific, we run GraphNAS over its own and EGAN's search space, given the same time budget (20 hours), and compare the final test accuracy of the searched architectures in Table 13. From Table 13, we can see that despite the simplicity of the search space, EGAN can obtain better or at least close accuracy compared to GraphNAS, which means better architectures can be obtained given the same time budget, thus demonstrating the efficacy of the designed search space.

### A.6.2 TEST ACCURACY DURING THE SEARCH PHASE

In this section, we compare the efficiency of EGAN and NAS baselines by showing the test accuracy w.r.t the running time, as shown in Figure 5, from which we can observe that the efficiency improvements are in orders of magnitude, which aligns with the experiments in previous one-shot NAS methods, like DARTS (Liu et al., 2019a).

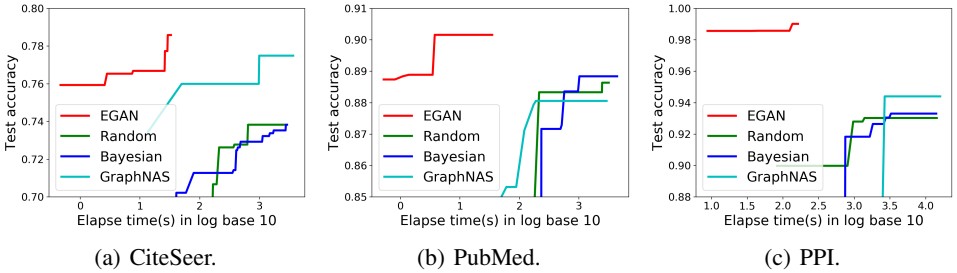

Figure 5: Test accuracy w.r.t. elapsed time on CiteSeer, PubMed, and PPI.

### A.6.3 MORE SEARCHED ARCHITECTURES

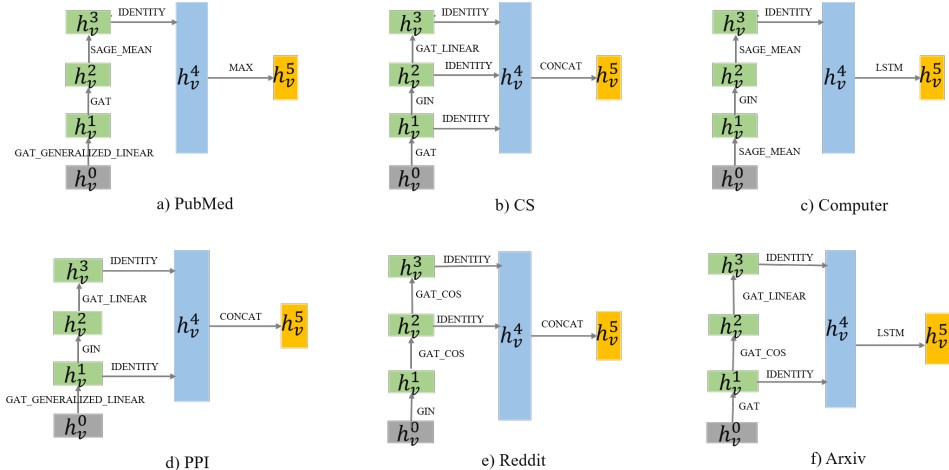

Figure 6: The searched architectures on more datasets in node-level tasks.

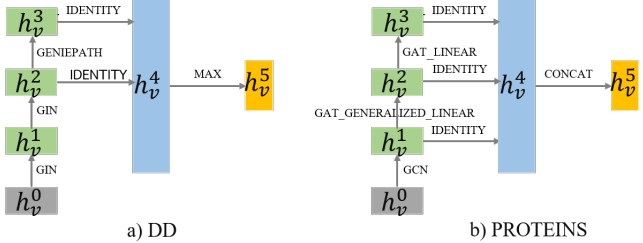

Figure 7: The searched architectures on datasets in graph-level tasks.

### A.7 MORE HYPERPARAMETERS

For all GNN baselines in node-level tasks, we use the $Adam$ optimizer, and set learning rate $lr = 0.005$, dropout $p = 0.5$, and $L_2$ norm to 0.0005. For other parameters, we do some tuning, and present the best ones in Table 14.

On Reddit, the parameters for GraphSAGE and GraphSAGE-JK are as follows: $lr = 0.005$, dropout $p = 0.5$, and $L_2$ norm to 0.0002, $K = 3, d = 64$, relu, epoch=30; For EGAN, $lr = 0.006$, dropout $p = 0.3$, and $L_2$ norm to 0.0005.

On Arxiv, the parameters of EGAN are as follows: $lr = 0.025$, dropout $p = 0.5$, and $L_2$ norm to 0.0005.

For all searched GNN architectures, the tuned hyperparameters are shown in Table 15.

Table 14: More implementing details of GNN baselines. Here we give *hidden dimension size, activation function*, and *the number of heads (GAT models)*. For JK-Network, we further give the layer aggregators.

| | Cora&CiteSeer | PubMed | CS | Computer | PPI | D&D | PROTEINS |
|---|---|---|---|---|---|---|---|
| GCN | 64, elu | 128, elu | 64, relu | 64, relu | 256, elu | 64, elu | 64, elu |
| GraphSAGE | 64, relu | 128, relu | 64, relu | 128, relu | 256, elu | 64, elu | 128, elu |
| GAT | 64, relu, 8 | 128, relu, 8 | 16, relu, 8 | 16, relu, 8 | 256, relu, 8 | 64, elu, 4 | 64, elu, 4 |
| GIN | 128, relu | 128, relu | 128, relu | 16, relu | 256, relu | 32, elu | 32, elu |
| LGCN | 128, relu | 128, relu | - | - | 256, relu | - | - |
| GeniePath | 256, tanh | 256, tanh | 128, tanh | 128, tanh | 256, tanh | 32, elu | 16, elu |
| JK-Network | CONCAT | CONCAT | CONCAT | CONCAT | LSTM | CONCAT | CONCAT |

Table 15: The hyperparameters obtained by hyperopt in the fine-tuning process for the searched architectures in Figure 2, 6, and 7. For all GAT aggregators, we set the number of head to 2 for simplicity, which empirically works well in our experiments.

| | Cora | CiteSeer | PubMed | CS | Computer | PPI | Reddit | Arxiv | D&D | PROTEINS |
|---|---|---|---|---|---|---|---|---|---|---|
| Hidden size | 32 | 64 | 256 | 256 | 64 | 1024 | 256 | 128 | 16 | 64 |
| Learning rate | 3.12e-3 | 5.937e-3 | 4.484-3 | 3.164e-3 | 2.111e-3 | 1.036e-3 | 5.440e-5 | 3.019e-3 | 3.867e-2 | 2.334e-2 |
| $L_2$ norm | 3.08e-5 | 2.007e-5 | 1.46e-4 | 2.46e-4 | 3.31e-4 | 0 | 1.495e-5 | 6.979e-5 | 2.16e-4 | 5.46e-4 |
| Activation function | elu | elu | elu | elu | elu | elu | relu | relu | elu | elu |
| Dropout rate | 0.4 | 0.7 | 0.6 | 0.6 | 0.5 | 0.5 | 0.4 | 0.4 | 0.3 | 0.3 |

## A.8 COMPARISONS WITH SNAS AND DARTS

In this section, to further demonstrate the efficiency of EGAN compared to DARTS (Liu et al., 2019a) and SNAS (Xie et al., 2019), we further record the trending of validation accuracy of the supernet by running them on the same search space during the search phase. These results are shown in Figure 8, from which we can see that EGAN can obtain larger validation accuracy more quickly than than SNAS and DARTS, which is attributed to the usage of natural gradient in Eq. (6).

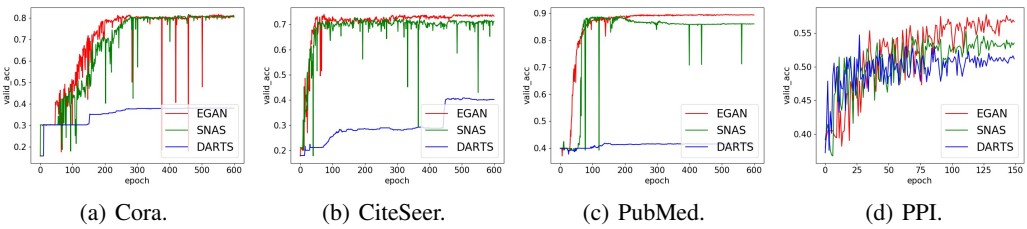

(a) Cora.  (b) CiteSeer.  (c) PubMed.  (d) PPI.

Figure 8: Validation accuracy w.r.t. elapsed time on CiteSeer, PubMed, and PPI.

