# OpenReview forum: "Efficient Graph Neural Architecture Search"
_ICLR.cc/2021/Conference — Reject_

### Official Review · AnonReviewer1 · 2020-10-22
**Efficient GNN-NAS approach, but might need more work**

**Rating:** 5
**Confidence:** 3

**Review:**

The paper presents a NAS method for graph-structured learning, which focuses on constructing a search space tailored to Graph Neural Networks (based on different node aggregators, skip-connections and layer aggregators).
In contrast to related GNN-NAS approaches, the authors apply a one-shot training paradigm, where model parameters and network architectures are learned simultaneously in an end-to-end fashion.
To enable NAS in large graphs, authors further apply a transfer learning scheme, where the optimal architecture is first learned on a sampled subgraph and model parameters are fine-tuned later on in the original graph.
The empirical evaluation includes experiments for node classification and graph classification, on which the proposed approach constantly performs better than or on par with human-designed GNNs, while being more efficient than related NAS approaches.

The paper is easily comprehensible, although it contains some typos and grammatical flaws.
The proposed approach is inspired by the idea to allow different GNN operators for different layers (adhering to different expressive capabilities), which is quite interesting and a fairly ignored topic in current literature.
In some sense, this is related to PNA [1], and it would be great to discuss this relationship in the paper.
The experiments look promising, showing both the strength of the proposed approach in terms of performance as well as in efficiency compared to related NAS approaches.

The training architecture and the model parameters are learned simultaneously (based on a Gumbel softmax formulation) via gradient descent.
As far as I understand, this requires the model to compute the output of every possible search state in every optimization step, which does not seem to scale to larger search spaces.
Is my understanding correct? And if so, how can the proposed method scale to larger search spaces?
Furthermore, other hyperparameters are tuned later on, e.g., dropout ratio or hidden feature dimensionality, which might prevent finding the "right" architecture in the first place.
Furthermore, this manual hyperparameter tuning might make the efficiency comparison to related NAS approaches somewhat unfair.

Additional comments:

* Eq. (4) refers to optimizing the network architecture based on training loss performance, while in general, one wants to find hyperparameters/network architectures that perform good on the validation set. Please clarify.

* Some baseline results are a bit weak and do not reflect the results reported in the official papers, e.g., for PPI and Reddit. For example, the GraphSAGE paper reports 95.4 micro F1 score on Reddit, while the paper reports 93.8.

* The transfer learning experiments do not seem that convincing to me (given my previous comment regarding Reddit results and the performance on ogbn-arxiv compared to a human-designed GNN). I personally think that the transfer learning idea has potential, but may need more work in order to see clear benefits.

* Is there some intuition why specific architectures win over others? Which architectures generally perform better than others? Can there be guidelines extracted for better GNN architecture design decisions?

[1] Corso et al.: Principal Neighbourhood Aggregation for Graph Nets (NeurIPS 2020)

=================== Post Rebuttal Comments ===================

I would like to thank the authors for their insightful rebuttal and clarifications.

Most of my concerns have been properly addressed, and I very much appreciate the discussion about PNA. However, since the authors train a "SuperNet", it is not particular clear to me why one even needs to decide for a specific aggregation scheme (in contrast to PNA), e.g., by simply using the softmax function instead of the gumbel softmax formulation.

Furthermore, I'm still not that convinced about the transfer learning proposal. In my opinion, a more in-depth analysis is needed (both theoretical and practical) to justify the claims made in the paper. Since GNNs do not even need to be trained in full-batch mode (i.e. via GraphSAGE, Cluster-GCN, GraphSAINT, ...), I'm not sure which benefits the proposed approach brings to the table in comparison to any other scalable methods.

Therefore, my rating will stay the same.

---

> ### Author Response · Authors · 2020-11-24
> **We update the paper based on your comments, and further answer your questions.**
>
> Thanks for your constructive reviews, and we have answered some common and important problems in a separate session, you may check it firstly, and other questions in the following.
>
>
> Q1. Discussion with PNA.
>
> Thanks for pointing this interesting work, and we have updated the discussion in the revised submission.
>
> In general, PNA proposes a composition of multiple aggregation functions in each GNN layer, while our framework searches for a single aggregation function in each GNN layer. Therefore, we can use our framework to help search for the combinations of multiple aggregation functions in a single GNN layer or include the PNA aggregator in our search space to see whether it can help further enhance the final performance.
>
> Q2. Gumbel-Softmax scale to large space.
>
> You are right. During the search, we still train the supernet, which means we need to compute the output of every possible search state. And the scalability problem does exist, especially large GPU memory consumption. In practice, EGAN with the current search space can work well on datasets like Cora (thousands of nodes) and PubMed (tens of thousands of nodes), while failing on datasets like Reddit or Arxiv (hundreds of thousands of nodes), which motivates us to design the transfer learning paradigm.
>
> In terms of scaling to larger search spaces, another way is to split the search space into different stages, like the hierarchical NAS in CV [1], which can be a future direction.
>
> Q3. Hyperparameters tuning.
>
> You are right. In terms of the fact that there exist performance gaps of the searched architecture in the search phase and retrain phase, including the hyperparameters tuning. Currently, in the NAS literature, it is a standard practice to do some hyperparameters tuning when retraining the searched architectures. Despite that the correlations between the performance in the search phase and the retrain phase are not proved in any paper, this pipeline empirically works well in practice. Therefore, we follow it and also find it works well in our experiments.
>
> In terms of the efficiency comparisons between NAS baselines and EGAN, we only compare the running cost during the search phase, since the searched architectures will be tuned with hyperopt for all NAS methods, thus the comparisons in Table 5 are fair actually.
>
> Q4. Optimizing based on training loss.
>
> You are right. In this work, we follow SNAS [2] to optimize the model parameters and network architectures based on training loss. In fact, optimizing network architectures, i.e., $\alpha$, in the training set and validation set, have been used in the NAS literature. Considering that the distributions of training and validation set usually are close and the hyperparameters of the searched architectures are tuned in the validation set, these two methods have little influence on the final performance in practice. In our experiments, we also observe this phenomenon, thus following the settings of SNAS.
>
> Q5. The results of GNN baselines are weaker than those in the original papers.
>
> Yes. As introduced in the paper, for fair comparisons, we use the implementations by PyG for all GNN baselines, and the data split ratios are 6:2:2(train/validation/test), which may be different from other papers, thus the performance of GNN baselines may be different from those in the original papers, however, we did try our best to tune all baselines with PyG to achieve the best performance we can. We will release the code later.
>
> In terms of GraphSAGE on Reddit, as introduced in Appendix A.4.4, the authors of JK-Network did not release their implementation, thus we use the implementation by the PyG framework. To keep it consistent, we also use the implementation for GraphSAGE by the PyG framework. Despite the performance of GraphSAGE is worse than that in the original paper, the trending of the performance gap between GraphSAGE and GraphSAGE-JK is the same as that the JK-Network paper, thus we report the performance as shown in Table 3.
>
> Q6. Transfer learning.
>
> We are sorry for the vagueness of the elaboration on the transfer learning. We have updated it in the revised version. And we also refer the reviewer to the reply to Q1 of Reviewer 4.
>
> Q7. Intuition about which architecture is better.
>
> This is a very interesting and open question. Currently, from the experimental results, we can observe that GAT tends to be selected, which demonstrates the general effectiveness of the attention mechanisms. In [3], the authors connect the properties of the graph (architecture) topology to the performance of neural networks but miss the analysis on the GNN architectures. I think this can be a very interesting and worthy direction to explore in the future.
>
> Refs.
> 1. Liu et al. Auto-DeepLab: Hierarchical Neural Architecture Search for Semantic Image Segmentation. CVPR 2019.
> 2. Xie et al. SNAS: STOCHASTIC NEURAL ARCHITECTURE SEARCH. ICLR 2019.
> 3. You et al. Graph Structure of Neural Networks. ICML 2020.

---

### Official Review · AnonReviewer3 · 2020-10-26
**We carefully review the motivation, approach, and empirical results.**

**Rating:** 3
**Confidence:** 5

**Review:**


This work proposes an efficient graph neural architecture search to address the problem of automatically designing GNN architecture for any graph-based task. Comparing with the existing NAS approaches for GNNs, the authors improves the search efficiency from the following three components: (1) a slim search space only consisting of the node aggregator, layer aggregator and skip connection; (2) a one-shot search algorithm, which is proposed in the previous NAS work; and (3) a transfer learning strategy, which searches architectures for large graphs via sampling proxy graphs. However, the current performance improvement over the human-designed models is marginal, which diminishes their research contribution.

The paper organization is clear, but some expressions should be improved. The details are listed as below.
Typos: In the Abstract, state-of-the-art should be abbreviated as SOTA, not SOAT.
Typos: $L_\theta(Z)$ after Equation (4) is not defined. Should it be $L_W (Z)$ as used in Equation (4)?
Clarity: The explanation before Equation (5) is a bit confused, which should be re-organized. There is grammar error (the absence of sentence subject) in the first sentence: “however, in this work, to make use of the differentiable nature of Lθ(Z), and design a differentiable search method to optimize Eq. (4).”
Clarity: The notations related to variable $Z_{i, j}$, i.e., $Z_{i, j}^T$ and $Z_{i, j}^k$,  are not defined well. What is the difference between the super-scripts: T and k?

The pros of this work are summarized in terms of three components used in EGAN, which improves the search efficiency. The experiment results show that their framework greatly reduce time, comparing with the GraphNAS, Bayesian search and random search.

Major questions:
(1) In Introduction: we doubt that designing proper GNN architectures will take tedious efforts. As far as I know, the architecture parameters of the human-designed models do not require extensive tuning efforts on the testing benchmark datasets. Furthermore, most of the architecture parameters could be shared and used among the testing datasets to achieve the competitive performances.
(2) It is unclear for the second challenge: the one-shot methods cannot be directly applied to the aforementioned dummy search space. There are some one-shot models with the parameter sharing strategy used for searching the hidden embedding size.
(3) In Section 3.1, why is the dummy search space very large? The search space seems only to include the aggregators and hidden dimensions. It might be much smaller than the search space of CNNs.
(4) Their search space assigns skip connections between the intermediate layers and the final layer, which is contradictory to the common case where the skip connections could be applied among the intermediate layers. As shown in [1], the skip connections may exist between any two layers. Could you provide reasons on the design of skip connection limitation?
(5) In the node and graph classification of the experimental section, the performance improvement over the human-designed is marginal. This would not justify the motivation of applying NAS to search graph neural networks. The authors should provide more discussions on the contribution of this work in terms of research and industrial applications.
(6) The marginal performance improvement might result from the search space. Currently, the authors’ search space is based on the traditional message passing approaches. They should consider more the recent developments in GNNs to further improve the performance.
(7) The selection of baselines is unfair. The search space contains the skip connection components based on the JK-Network. However, authors excluded the important baseline in [2], which could achieve the comparable performance on dataset Citeseer and Reddit. For the graph classification task, authors also excluded a lot of pooling methods, such as the Graph-u-Net [3], which achieves the better performance than the proposed approach.

[1] Rong, Yu, et al. "Dropedge: Towards deep graph convolutional networks on node classification." International Conference on Learning Representations. 2019.
[2] Xu, Keyulu, et al. "Representation learning on graphs with jumping knowledge networks." arXiv preprint arXiv:1806.03536 (2018).
[3] Gao, Hongyang, and Shuiwang Ji. "Graph u-nets." arXiv preprint arXiv:1905.05178 (2019)

---

> ### Author Response · Authors · 2020-11-24
> **We carefully reply for the motivation, approach, and experiments, and update the paper accordingly.**
>
> Thanks for your constructive reviews, and we have answered some common and important problems in a separate session, you may check it firstly, and other questions in the following.
>
> Q1. Typos and clarity.
>
> We have revised the paper accordingly in the revised submission.
>
> Q2. The necessity of NAS for GNN in the introduction.
>
> To the best of us, we do not find the claim in any paper that "most of the architecture parameters could be shared and used", while in a recently released paper [1] in NeurIPS 2020, the authors show that the architectures can transfer across different tasks when they are similar in terms of the task space (see more details in [1]).
>
> On the other hand, we agree that "tedious effort" may be kind of exaggerative in designing data-specific GNN architectures, considering that the GNNs are not that complex as the CNNs. Thus we remove "tedious" in the revised submission.
>
> Moreover, as mentioned in [1], there are other tasks, like circuit design, SAT generation, subgraph matching, which can be quite different from the widely-seen node and graph classification tasks, thus more efforts have been spent on these tasks. In this sense, our framework can be applicable to more tasks, which can be future directions of our work.
>
> Q3. The large dummy search space, and one-shot method to search for hidden size.
>
> In terms of the dummy search space including all model parameters, they are not limited to the hidden embedding size and aggregator functions but include other parameters, like the activation function, attention heads, number of layers, learning rate, batch norm, optimizer, etc. Then the Cartesian product of all these parameters leads to a large search space, which can be computationally expensive for a NAS method.
>
> In terms of the second challenge, thanks for pointing out that one-shot NAS methods can be used to searching for hidden embedding size. We made a mistake here, and have updated the paper accordingly.
>
> Q4. Searching for hidden size && skip connections between intermediate layers.
>
> These two choices can be easily integrated into the search space of EGAN. However, in this work, we focus on the aggregation functions, which are related to the GNN expressive capability, and leave these two choices for future work.
>
> Q5. Performance gain is marginal.
>
> We refer the reviewer to the reply to Q2 of Reviewer 4 and emphasize that the performance gains are sense-making.
>
> Q6. Performance gains are from the search space && include more recent baselines.
>
> Firstly, we refer the reviewer to the reply to Q3 of Reviewer 2, which emphasizes the contribution of the search space.
>
> We construct our search space on top of the message passing framework, since the most popular and widely-used GNNs, e.g., GCN, GAT, GraphSAGE, are relying on it, and as mentioned in the reply to Q4, one of the purpose is to show the effectiveness of NAS for GNN, thus the contributions of the proposed EGAN.
>
> For other models beyond the message passing framework, e.g, GEOM-GCN, or Positional-GNN [2], we leave them for future work.
>
> Q8. Baselines are not fair && JK-Network && Graph-U-Net.
>
> In terms of the JK-network, in fact, as introduced in the paragraphs following the GNN baselines in Appendix A.3.2, we implement the JK versions for GCN, GraphSAGE, GAT, GIN, and GeniePath, and only report the better performance of the original models and JK versions. To make this clear, we give the detailed results of the 10 models in Appendix A.4.5 and A.5.3 in the revised submission, and for the sake of space limitation, we keep Table 2&4 unchanged.
>
> In terms of Graph-U-Net, which is an advanced pooling method on top of the encoder-decoder framework, while our method adopts a simple pooling method (summation) on top of the searched GNN architectures, thus it might be possible that Graph-U-Net outperforms EGAN in the pooling related tasks, and we leave as future work searching for aggregation and pooling methods in GNN.
>
> Refs.
> 1. You et al. Design Space for Graph Neural Networks. NeurIPS 2020.
> 2. You et al. Position-aware graph neural networks. ICML 2019.

---

### Official Review · AnonReviewer2 · 2020-10-27
**Concerns about the novelty and the experimental design**

**Rating:** 5
**Confidence:** 5

**Review:**

This paper presents a differentiable NAS method named EGAN for automatically designing GNN architectures. The main contribution is searching GNN architectures efficiently with an one-shot framework based on stochastic relaxation and natural gradient method. Extensive experiments conducted on node-level and graph-level tasks show the efficiency and effectiveness of the proposed search method.

Pros:

+ Paper is well-written and easy to follow;
+ The proposed search space with node aggregators, layer aggregators is interesting;
+ The design of the baseline methods including random and bayesian search is appreciated;
+ Empirical results on different datasets and tasks are very strong.

Cons:

- Limited novelty, the proposed search method is very similar to SNAS (Xie et al., 2018) except the search space;
- A similar one-shot search method for GNN has been proposed in SGAS (Li et al., 2020) which weakens the claimed contribution of being the first one-shot NAS method for GNNs;
- It is not clear why stochastic natural gradient method is needed;
- The performance of GraphNAS with EGAN’s search space (Table 11) is close to the performance EGAN (Table 2). Therefore, the performance gain mainly comes from the well-designed search space;
- There is a lack of comparison of the models' parameters across different search methods. Thus, it is not clear whether the experiments are conducted under a fair setting.

Other Comments:

* The obtained architecture in Figure 2 b) includes SAGE. But it is not clear which aggregator is using;
* The contribution in terms of transfer learning is weak since the proxy graph is a subsample of the large graph. The identical nodes have been exposed during the search phase.

References:
* Xie, S., Zheng, H., Liu, C. and Lin, L., 2018, September. SNAS: stochastic neural architecture search. In International Conference on Learning Representations.
* Li, G., Qian, G., Delgadillo, I.C., Muller, M., Thabet, A. and Ghanem, B., 2020. Sgas: Sequential greedy architecture search. In Proceedings of the IEEE/CVF Conference on Computer Vision and Pattern Recognition (pp. 1620-1630).

Post rebuttal Comments:
Thank the authors for the detailed response. I keep my rating as 5.

---

> ### Author Response · Authors · 2020-11-24
> **We updated the paper for more experimental details and further clarify the novelty here.**
>
> Thanks for your constructive reviews, and we have answered some common and important problems in a separate session, you may check it firstly, and other questions in the following.
>
> Q1. Limited novelty because of SNAS && Why stochastic natural gradient is needed.
>
> We agree with you that the stochastic relaxation method is similar to that in SNAS, which uses the Gumbel-Softmax. However, the optimization algorithm is different from the one used in SNAS. We build the optimization method on top of the natural gradient descent for NAS in [2], which is shown to provide more robust training processing.
>
> In the revised version, we add one section in Appendix A.8 to compare the performance of EGAN with DARTS[5] and SNAS on the same search space to show the advantages of our method, especially over DARTS [5].
>
> Q2. Not the first one-shot NAS.
>
> Thanks for providing this relevant works. Indeed, SGAS is a novel one-shot method, which shows some promising results in GCN architecture search as an auxiliary task, since they only use one benchmark dataset, i.e., Reddit, in the conventional GNN based tasks. Besides SGAS, another work [3] in AAAI 2020 has designed a one-shot framework to search for GCN architectures in human action recognition.
>
> However, in our paper, we conduct systematic and comprehensive studies on the GNN architecture search problem, which are evaluated in extensive (10) datasets in the conventional node-level and graph-level classification tasks, the most important two in GNN literature.[4]. In this sense, to the best of our knowledge, our work is the first one-shot NAS framework for the conventional GNN tasks. We have added SGAS and [3] in the related works in the revised submission.
>
> Q3. Performance gain of search space.
>
> You are right. With the search space of EGAN, GraphNAS can obtain competitive performance with EGAN. This clearly demonstrates the superiority of the search space, which is the first contribution of the proposed work.
>
> Besides the better performance than GraphNAS, EGAN is two orders of magnitude more efficient than GraphNAS, as shown in Table 5, which is the second contribution of our work.
>
> Q5. Compared to different search methods.
>
> If my understanding is right, you want to know the hyperparameters of baselines. We have added one section to introduce the hyperparameters in Appendix A.7 in the revised submission.
>
> Q6. SAGE in the cases.
>
> Thanks for pointing out this. We have updated Figure 2&6 accordingly.
>
> Q7. Identical nodes in the sampled graphs.
>
> You are right. The nodes that occurred in the sampled graphs can benefit from the supernet training. However, the proportion of the sampled nodes is 15%, while the remaining 85% nodes do not occur in the sampled graph. Therefore, we believe the contribution of the transfer paradigm still makes sense in enabling the architecture search in large graphs.
>
> Refs.
>
> 1. Jure et al. Sampling from Large Graphs. SIGKDD 2006.
> 2. Akimoto et al. Adaptive Stochastic Natural Gradient Method for One-Shot Neural Architecture Search. ICML 2019
> 3. Peng et al. Learning Graph Convolutional Network for Skeleton-based Human Action Recognition by Neural Searching. AAAI 2020.
> 4. Hu et al. Open Graph Benchmark: Datasets for Machine Learning on Graphs. NeurIPS 2020.
> 5. Liu et al. DARTS: DIFFERENTIABLE ARCHITECTURE SEARCH. ICLR 2019.

---

### Official Review · AnonReviewer4 · 2020-10-28
**Official Blind Review #4**

**Rating:** 5
**Confidence:** 4

**Review:**

Summary: The paper proposes a framework for efficient architecture search for graphs. This is done by combining a differentiable DARTS-like architecture encoding with a transfer learning method, that searches on smaller graphs with similar properties, and then transfers to the target graphs. The experiments show that EGAN matches or exceeds both hand-designed and NAS-designed GNNs. Moreover, the method is very fast to run.

Recommendation: Overall, I am voting to reject, as several crucial pieces of information are missing from the current draft, and some other parts are unclear. The most novel part of the paper seems to be the transfer of architectures learned on subgraphs to larger graphs, but there is little discussion on how that impacts downstream accuracy, or if the transfer learning is at all needed given that the method is already very efficient. Moreover, there is no information on how the choice of method used to select subgraphs affects the entire framework. The experimental results look promising, but there should be more care taken to assess statistical significance.

Main pros:
1. The general concept of adapting ideas from DARTS to work with Graph Neural Networks is fairly natural.
2. The set of tasks that are considered is broad, and the comparison is performed across a range of baselines.
3. Selecting subgraphs with similar properties to the full graph, searching for good architectures on those, and then transferring to the full task, is a very interesting idea.

Main cons:
1. The transfer learning method is only described very briefly, which leaves several open questions
- How much are the graphs reduced? Section 3.3 mentions 15% of original size, but it's unclear if that concerns the framework presented in the paper, or is that a value advocated for in some related work. It is also unclear if that refers to number of nodes or number of edges; surrounding text seems to suggest the former, but then the GNN running time will be more affected by the latter.
- What is the empirical impact of the reduced graph size on running time?
- As the proposed method seems to be very fast, is it unfeasible to run the search on the full graphs? This would allow to estimate how much accuracy is lost due to searching on a proxy task instead of using the target task, and how that changes as one varies the graph sizes for the proxy task.
- The paper mentions using Random PageRank Node to select the subgraphs; I wonder how that choice affects the results as opposed to doing something more trivial (e.g. dropping nodes and/or edges uniformly at random).
2. The bolding of results (Tables 2, 4, 6) is a bit misleading, since many of the differences do not seem statistically significant (e.g. Computer, Proteins) or are even zero (D&D). It would be better to perform a statistical significance test, and make the statistical significance clear by bolding several joint top values.
3. From Appendix 3.2, I understand that most baseline results were produced by the authors (as opposed to copying the numbers from related works). How were the hyperparameters for the different baselines tuned? In particular, I'm concerned about the last paragraph on page 14, which mentions that the number of layers for all models was fixed to 3. As the number of layers is one of the most important GNN hyperparameters, I'm not sure if simply fixing it to 3 for all baselines is entirely fair.

Other comments:
- The paper says that the output of the last GNN layer is always used by the layer aggregator, motivating it as injecting relational bias. My understanding is that this just ensures there are no "dead ends" in the resulting computational graph, which is a common idea in NAS works; I'm not sure how relational bias is related to this.
- [1] defines two variants of the Gumbel Softmax trick: besides the basic one, there's also the straight-through variant, which uses hard (i.e. one hot) samples on the forward pass, and the soft approximations (as defined in Equation 5) on the backward pass. Which variant did the authors use?
- The paper motivates the transfer learning approach by saying that a k-layer GNN needs to process k-hop neighbours, the number of which can be exponential in k. This seems to suggest the GNN running time grows exponentially with k, which is of course not true; in fact, every GNN propagation step requires the same amount of compute, proportional to the number of nodes and edges in the graph.
- The first dot in Section 3.4 says that EGAN is better than existing GNN-NAS methods, as it has a larger (and thus more expressive) search space, but then goes on to say that EGAN is also better as it has a smaller (and thus cheaper to search) search space. This feels a bit contradictory; it would be fine to just state that the EGAN search space has a different focus (i.e. searching only over the "core GNN propagation + aggregation" part).
- [2] report several simple (hand-designed) GNN variants that get 0.992 on PPI.
- Table 3 reports that searching on the Arxiv dataset took 10 seconds. How is that even possible? As far as I understand, the architecture search involves training a supernet (containing 12 different aggregation types for each of the 3 GNN layers, among other things) for some number of epochs, repeating that process 5 times for different seeds, and then retraining the best architecture from scratch. Can you comment on how long each of these stages takes?
- In the GNN literature, results on D&D and Proteins (Table 4) are reported in two different ways: some papers (e.g. [3]) report the validation set results as the final metric (despite it being available to the model selection procedure), while others (e.g. [4]) report the result on the test set (which was not seen by any part of the pipeline). I understand the authors follow the latter strategy - can you confirm?
- Appendix 4.2 says "we use the global sum pooling method at the end of retraining" - what does that mean?
- Reading the paper feels a bit bumpy, since there are some sentences that are hard to read, and therefore could be revised. Examples (I include only part of each sentence, just to make it identifiable):
  - Page 2: "In the literature, we (...)"
  - Page 4: "Note that in our work (...)"
  - Page 6: "Jiang & Balaprakash (2020) is (...)"
  - Page 7: "First,, we (...)", "Second, we (...)"
  - Caption of Table 5
  - Page 16: "For other methods, (...)"

Small remarks, typos, and grammar issues (did not influence my rating recommendation):
- The EGAN abbreviation may be a bit misleading, since one could assume it refers to Generative Adversarial Networks
- Abstract: "SOAT"
- Page 1: "one shot methods NAS methods"
- Page 1: "in orders more efficient" -> "orders of magnitude more efficient"
- Pages 1 & 8: missing space after citation
- Pages 2 & 3: "for the best of our knowledge" -> "to the best of our knowledge"
- Page 3: "computational" -> "computationally"
- Equation 1: l should be in parenthesis
- Page 3: "have been demonstrated effectiveness" -> "have been demonstrated effective"
- Page 4: "kth" -> "k-th"
- Page 5: "To robust the final results" -> "To make the final results more robust"
- Page 8: "BY"
- Page 13: "feature vector obtains" -> "feature vector obtained"
- Page 16: "as evaluate metric" -> "as the evaluation metric"
- Throughout the paper: "In this part" -> "In this section"

References:
- [1] Categorical Reparameterization with Gumbel-Softmax
- [2] GNN-FiLM: Graph Neural Networks with Feature-wise Linear Modulation
- [3] How Powerful are Graph Neural Networks?
- [4] A Fair Comparison of Graph Neural Networks for Graph Classification

----------------------------------------------------------------------------------------------------

Comments after rebuttal:

I would like to thank the authors for their detailed response. Many things were addressed, and I have increased my score to 5 to reflect that the paper is not far from acceptance threshold. I think the main thing missing is a discussion of the effect of the sampling of subgraphs: i.e. showing that PageRank is indeed better than choosing nodes at random, and analysing how the results change when the reduction percentage is varied (between a low value and the maximum value that fits in GPU memory).

---

> ### Author Response · Authors · 2020-11-24
> **Thanks for your constructive and comprehensive comments, and we reply to them one by one.(1/2)**
>
> Thanks for your constructive and comprehensive reviews, we really appreciate it.  Here have answered some common and important problems in a separate session, you may check it firstly, and other questions in the following.
>
> Q1. More technical details of transfer learning
>
> a) Sampling size and methods.
>
> The used sampling method, i.e., RPN, and size are actually following an established work in [2], which shows that sampling 15% of the total nodes based on the PageRank values works best to preserve the static and evolutionary graph patterns, compared to other sampling methods, e.g., random node or edge sampling methods.
>
> In our experiments, we show that this sampling strategy, i.e., randomly sampling 15% nodes based on PageRank values to construct the proxy graph, empirically works well. We agree that it will better if we can show the comparisons of different sampling methods and ratios, and we will add these experiments in future.
>
> b) Empirical time reduce && directly search on the full graph.
>
> We are sorry for the missing information about the transfer experiments in the main part of the paper, which are partially introduced in Appendix A 3.4. Here we further clarify the following points:
>
> - For the results of the last row in Table 3, i.e., the results of EGAN. The search process is conducted on the smaller versions of these two datasets, recall the 15% nodes of the graphs, and the corresponding performance (Micro-F1 and Accuracy) are evaluated on the original graphs with the transferred architectures.
> - Therefore, we can see that the performance of EGAN in transfer settings can be guaranteed. On Reddit, it outperforms all baselines, and on Arxiv, the performance is close to the best GNN model. It also empirically demonstrate that sampling 15% nodes from the original graph empirically works well.
> - In terms of directly searching on the full graph, note that in Table 3, NAS baselines are executed directly on the full graphs, which leads to the OOM problem of running the released code of GraphNAS. Moreover, when comparing the search cost, we return the best architecture after running the NAS baselines after 72 hours, while the search cost of running EGAN on the sampled graphs are in seconds. Note that for the transfer learning paradigm, we need to spend some time in preprocessing, including computing the PageRank of the graph, and sampling the small graph, which costs less than one CPU hour on Reddit and Arxiv.
> - In terms of running EGAN directly on the full graphs, currently, our GPU memory is not enough to train the supernet. This can also demonstrate the difficulties in conducting architecture search in large graphs, even adopting the one-shot NAS method.
>
> Q2. A statistically significant test of the performance gain.
>
> Firstly, we emphasize here the improvements of EGAN are on par with existing works, like those reported in GAT[3], Geom-GCN[4], and DropEdge[5], thus the performance gains are sense-making. Second, on D&D, the mean accuracy of EGAN and DiffPool are the same, since DiffPool designs a hierarchical pooling method on top of the GCN, while our method adopts a simple pooling method (summation) on the searched GNN architectures, which actually demonstrates the effectiveness of the searched GNN architecture. Thirdly, we conduct a significance test (p < 0.05) for the results in Table 2&4, which is statistically significant.
>
> Q3. Hyperparameters of baselines. K=3 seems not good for baselines;
>
> We have updated the hyperparameters of all baselines in Appendix A.7 in the revised submission. We set the number of layers to 3 for all GNN baselines, since the search space of EGAN is on top of 3-layer GNN.
>
> Q4. Relational bias in the layer aggregators.
>
> Firstly, we are sorry for the inaccurate terms, which should be relational inductive bias from the positional paper[6]. Secondly, generally speaking, we agree with you that this is a common idea in NAS. Thirdly, in GNN with layer aggregators, it means to guarantee K-hop neighborhood access by adding this constrain, which we regard as a kind of relational inductive bias. However, to avoid misunderstanding, we changed the description in the revised submission.
>
> Q5. Variant of Gumbel-Softmax.
>
> We use soft approximation.
>
> Q6. Contradictory of large and small search space.
>
> We thank you for the constructive comment and have revised the paper accordingly.
>
> Refs.
> 1. Hamilton et al. Inductive Representation Learning on Large Graphs. NeurIPS 2017.
> 2. Jure et al. Sampling from Large Graphs. SIGKDD 2006.
> 3. Velickovic et al. Graph Attention Networks. ICLR 2018.
> 4. Pei et al. GEOM-GCN: GEOMETRIC GRAPH CONVOLUTIONAL NETWORKS. ICLR 2020.
> 5. Rong et al. DROPEDGE: TOWARDS DEEP GRAPH CONVOLU- TIONAL NETWORKS ON NODE CLASSIFICATION. ICLR 2020.
> 6. Battaglia et al. Relational inductive biases, deep learning, and graph networks. arxiv 2018.

---

> > ### Author Response · Authors · 2020-11-24
> > **Thanks for your constructive and comprehensive comments, and we reply to them one by one.(2/2)**
> >
> > Q7. Another baseline achieves 0.992 on PPI.
> >
> > Thanks for pointing out this interesting paper, and in fact, there are other works that can achieve 0.995 on PPI. (check https://paperswithcode.com/sota/node-classification-on-ppi). However, in this work, we build our search space on the message passing framework and focus on the node and layer aggregators, therefore, we compare EGAN with existing GNN models regarding aggregation functions, the results in Table 2, 3, and 4 show that the proposed method can achieve better or competitive results on various tasks, not only one dataset, by searching for data-specific GNN architectures.
> >
> > Moreover, the paper you mentioned here is focusing on feature processing beyond the aggregation function, therefore can be integrated to further improve the performance of EGAN, which we leave as future work.
> >
> > Q8. 10 seconds in Arxiv
> >
> > As explained in the replies of Q1(b), it is the search process, i.e., the training of the supernet, on the smaller version of the Arixv dataset that costs 10s. This time is usually reported as the search cost in the NAS literature. For retraining the best architecture from scratch, we use hyperopt to do the hyperparameters tuning, the steps of hyperopt are set to 50, and the average cost is around half an hour.
> >
> > Q9. Validation or test on D&D and Protein
> >
> > We report the results on the test set.
> >
> > Q10. Global sum pooling.
> >
> > We are sorry for the vagueness. The global sum pooling means represents that get the summation of the embeddings of all nodes to represent the whole graph embedding, i.e., 	$z = \sum_{i \in \mathcal{V}} h^{l}_i$.
> >
> > Q11. Writings and typos.
> >
> > We thank you for pointing out these problems and have updated the paper.
> >
> > Refs.
> >
> > 1. Hamilton et al. Inductive Representation Learning on Large Graphs. NeurIPS 2017.
> > 2. Jure et al. Sampling from Large Graphs. SIGKDD 2006.
> > 3. Velickovic et al. Graph Attention Networks. ICLR 2018.
> > 4. Pei et al. GEOM-GCN: GEOMETRIC GRAPH CONVOLUTIONAL NETWORKS. ICLR 2020.
> > 5. Rong et al. DROPEDGE: TOWARDS DEEP GRAPH CONVOLU- TIONAL NETWORKS ON NODE CLASSIFICATION. ICLR 2020.
> > 6. Battaglia et al. Relational inductive biases, deep learning, and graph networks. arxiv 2018.

---

### Author Response · Authors · 2020-11-24
**We answer the common questions to all reviewers.**

### To all reviewers

Thank you all for the constructive reviews. We have revised the paper and submitted the latest draft with the revised part in blue.

Here we firstly summarize the key points you all have concerns about and then reply to other questions accordingly.

Q1. The motivation and novelty in architecture search for GNN.

In general, it is a meaningful task to automatically design data-specific GNN architectures, since GNNs have been used in diverse domains, including recommendation, fraud detection, circuit design, SAT generation, subgraph matching, etc., as introduced in [1]. These tasks usually correspond to graphs of different properties, thus the performance of existing GNNs vary (see more discussions in [1]). This leads to a huge demand for automatically designing GNN architectures, especially for novel tasks like circuit design.

Secondly, it leads to quite a large search space when including every aspect into the search space. As introduced in [1], there are 315,000 GNN architectures when simply including 12 aspects. Considering their choices are not exhaustive, for example, they only consider 3 node aggregators (max, min, sum), while we include 12 node aggregators. Then the dummy search space mentioned in our paper can be far larger, which makes it valuable to simplify the search space and design more efficient search algorithms for architecture search in GNN. Thus we can see that the first two contributions of EGAN, i.e., the expressive search space focusing on aggregation functions and the one-shot search algorithm, are well motivated and quite important.

Thirdly, the computational cost, especially the GPU memory consumption, becomes extremely expensive in large graphs as mentioned in [2], which is a unique challenge in searching for GNN architectures. This motivates us to design the transfer learning paradigm to enable architecture search in large graphs.

To summarize, it is an important and challenging problem to conduct architecture search for GNN, for which the proposed framework in this work provides a novel, effective, and efficient way.

Q2. The motivation for transfer learning

As introduced in [2,3], when training GNNs in large graphs, in each batch, the time and memory cost increase exponentially w.r.t. K, i.e., the number of GNN layers, with the worst cases of $O(|\mathcal{V}|)$. Obviously, it is extremely expensive in large graphs for any GNN models, and [1] uniformly samples a fixed-size set of neighbors to train a scalable GNN model. The situation becomes more severe when conducting architecture search in large graphs since we are training a supernet including various GNN models.
To address the above problem and enable architecture search in large graphs, we propose the transfer learning pipeline to firstly search for GNN architectures in proxy (small) graphs, which means $O(|\mathcal{V}|)$ is not a big challenge for us to train the supernet, and then transfer the searched GNN architecture in large graphs. We argue that this is a meaningful manner to enable architecture search in large graphs, which broadens the applications of the proposed framework.

Q3. Including more GNN models.

Firstly, we construct our search space on top of the message passing framework, since the most popular and widely-used GNNs, e.g., GCN, GAT, GraphSAGE, are relying on it, and we specifically focus on the aggregation functions. Considering that one of the purposes is to show the effectiveness of NAS for GNN, not to design a perfect method to include all GNN models, currently, we do not include GNN models beyond the message passing framework.

In Appendix A.3 in the revised draft, we add one section to discuss more the comparisons between EGAN and other GNN models, including the PNA mentioned by Reviewer 1.

Refs.
1. You et al. Design Space for Graph Neural Networks. NeurIPS 2020.
2. Hamilton et al. Inductive Representation Learning on Large Graphs. NeurIPS 2017.
3. Jia et al. Redundancy-Free Computation for Graph Neural Networks. KDD 2020

---

### Decision · Program_Chairs · 2021-01-07
**Final Decision**

**Decision:**

Reject

**Comment:**

This paper presents a differentiable neural architecture search method for GNNs using Gumbel softmax-based gating for fast search. It also introduces a transfer technique to search architectures on smaller graphs with similar properties as the target graph dataset. The paper further introduces a search space based on GNNs message aggregators, skip connections, and layer aggregators. Results are presented on several undirected graph datasets without edge features on both node and graph classification.

The reviewers mention that the results are promising, but they unanimously agree that the paper does not meet the bar for acceptance in its current form. I tend to agree with the reviewers in that the effect of the individual contributions (search space vs. method vs. transfer) needs to be better disentangled and studied independently, and that it is unclear why selecting a single aggregation function out of many is important vs. choosing multiple ones at the same time such as in PNA [1] as pointed out by R1. This should be carefully studied going forward. Lastly, all reviewers agreed that the proposed transfer method requires more detailed experimental validation and motivation.

[1] Corso et al.: Principal Neighbourhood Aggregation for Graph Nets (NeurIPS 2020)